# OpenReview forum: "Intrinsic Evaluation of Unlearning Using Parametric Knowledge Traces"
_ICLR.cc/2025/Conference — Submitted to ICLR 2025_

### Official Review · Reviewer_Kjdc · 2024-10-29

**Soundness:** 3
**Presentation:** 4
**Contribution:** 4
**Rating:** 8
**Confidence:** 4

**Summary:**

Developers sometimes wish to remove learned knowledge from a Large Language Model (LLM), which is a task known as unlearning. This work presents a new benchmark for evaluating unlearning methods, one which performs internal checks for unwanted knowledge (stored in the model parameters) in addition to the usual behavioural checks. This is made possible by the systematic detection of ‘concept vectors’ within LLMs, via the projection of weights into the vocabulary space and some causal testing. To justify the methodology of the benchmark, the authors provide a baseline method that is actually able to erase these ‘concept vectors’, resulting in a minimal change to the model that has a large impact on the unwanted behaviour. Furthermore, the authors demonstrate that current unlearning methods are unable to fully remove these ‘concept vectors’ (without impacting unrelated concepts) and this results in a vulnerability to adversarial attacks that can exploit these residual parameters.

**Strengths:**

+ This work is of high quality, making numerous contributions in one paper. The authors not only present a novel benchmark (leveraging a fairly novel interpretability finding) but they also take the next step of evaluating popular unlearning methods and a proof-of-concept baseline that is more suited to the benchline. Cleverly, this allows them to justify that success on this benchmark does track with desirable properties like specificity and attack robustness. This is significant for any future unlearning work.

+ The authors also do a good job in clarity and presentation, segmenting the paper into cohesive sections, as well as making good use of diagrams and examples. Formalisation through maths notation is also helpful.

**Weaknesses:**

+ The baseline method ‘Needle’ scores well on intrinsic evaluations while performing badly on behavioural evaluations, which gives some worry that the former could be maximised without providing what we actually want (the change in behaviour). The expected response is that this is why both evaluations are necessary, or the fact that the ‘Needle’ intervention was very small compared to the other methods - but this could be better emphasised by keeping the size of change fixed between methods.

+ There are some other potential worries with the benchmark. Since benchmark concepts are chosen based on whether a corresponding ‘concept vector’ exists, this raises the question whether those concepts are more easily unlearned than the alternatives (which may be stored more indirectly/ inaccessibly in the model). Future work that is able to create something like ‘Needle’ without assuming knowledge of the evaluation concept vectors should evaluate against an external benchmark to see whether there is a concept selection bias.

+ As the authors have pointed out, a concept/ behaviour could be defined by multiple ‘concept vectors’ and model features in general. There may be some benefit to lumping together similar found vectors, for example, you may be evaluating elimination of two similar vectors separately and a given method could be doing a great job in eliminating only the intended concept vector but it is the opposite one from the one being evaluated? More generally, the authors method starts with the most interpretable vectors and then keeps them if they are causally effective, but it could be better to prioritise causal effectiveness first since unlearning these vectors is what we actually care about (though would make construction of the benchmark more difficult).

+ In terms of adversarial attacks, the previous point is irrelevant, since any remaining ‘concept vector’ could be found and exploited. However, if protection against adversarial attacks is what we really want, then it seems simpler and more robust to have the benchmark/ methodology just be the evaluation against adversarial attacks - more robust because it does not rely as much on finding every vector for a concept.

+ The adversarial attack experiments in section 5 are excellent and convincing, however this section would greatly benefit from using a sample size greater than 10. Additionally, ‘Needle’ could have been added to Table 4 and LRLs results should be analysed more thoroughly since they are so surprising.

**Questions:**

+ Have the points I have raised in ‘weaknesses’ already been considered by you? Do you believe them to be valid criticisms?

+ In step 1 of section 3.1, tou mention that many found concepts were “syntactic (e.g. plural verbs)”. How do you unlearn or test for unlearning something like plural nouns? Is this something ever desirable to unlearn? Wuestions and text completions to do with the theory of plural nouns does not seem relevant to a feature which lights up for all plural nouns, so are the questions more to do with asking whether a word is a plural noun?

+ Is there any reason for naming the matrices W_k and W_v in section 2? This is usually what you call the key and value weight matrices in the attention mechanism and could be confusing.

+ Is vocabulary projection better at finding concept vectors than just promoting/ ablating them and then seeing the causal impact on token prediction?

---

> ### Author Response · Authors · 2024-11-21
> **Responses to the Reviewer Kjdc**
>
> We thank the reviewer for their favorable review and insightful comments. We are encouraged the reviewer appreciates our contributions and sees the merits of our work.
>
> **Regarding the weakness points raised:**
>
> **W1:** Thanks for this thoughtful comment and suggestion. We will add an explicit specification of the amount of parameters modified by each method. Regarding maximizing the intrinsic evaluation without changes in the model’s behavior – this is an important point. We agree with the reviewer that a complete erasure of the concept vector does not provide the fully desirable behavioral effect (e.g. very low BLUE score, full resistance to jailbreak attacks). However, given the very low performance of current unlearning methods on ConceptVectors, we believe it is a strong benchmark that makes the first step toward parametric removal.
>
> **W2:** Yes, our data selection process focuses on vectors that are easily located, very disentangled and show clear concepts that are causally related to the concept, which may be easier to remove. However, we see great value in these vectors for evaluation, as our empirical results show that even in these cases, unlearning methods perform badly in terms of removing parametric knowledge, showing there is a large room for improvement of current unlearning methods. Using less prominent vectors could be a valuable direction for future work to explore.
>
> **W3:** Thanks for this comment. The reviewer’s idea of prioritising causal effectiveness first and then interpretability is intriguing, though it will require changing the whole data generation process. We would also like to point out that this approach has a risk – we do not know in advance which vectors are polysemantic and capture multiple concepts in superposition, but this is expected to be prevalent based on prior findings. Consequently, we may need to explore many concepts before finding those that their interpretation is very clear. In that sense, we believe our approach is more direct for locating these specific concept vectors.
>
> **W4:** We agree that evaluation that is based on adversarial attacks is valuable as well, though similarly to behavioral evaluations, it relies on the model’s outputs without directly targeting the source of the knowledge.
>
> **W5:** We thanks the reviewer for appreciating our experiments. We are happy to add more samples to these experiments and doing this analysis of the LRL attack, and will try to do so during the next few days. However, given that we have received 7 reviews, we may not make it on time.
>
> **Regarding the reviewer’s questions:**
>
> **Q1:** Thank you for raising all these thoughtful points. Please see our responses to W1-W5 above, where we addressed them.
>
> **Q2:** As explained in lines 188-190 (Section 3.1), we do not include lexical or syntactic concepts. Our focus is on semantic concrete concepts that the model would capture facts about.
>
> **Q3:** Previous work showed that the MLP layers in transformer-based language models can be cast as key-value memories. We followed this notation.
>
> **Q4:** Thanks for this great question. We haven’t made this kind of comparison and are not aware to other works that did. There are two important things to note here: First, in terms of computation and evaluation, localizing concept vectors through ablations is substantially more expensive and complex than vocabulary projections. It requires executing the model for every vector, crafting inputs that could be used for testing the possible encoded concepts, and measuring the effect on the output is not always trivial. Vocabulary projections provide a simpler and more direct way to identify the encoded concepts. The second thing to note is that practically we do both, as step 3 in our pipeline causally validates the concept identified with vocabulary projection.
>
> We thank the reviewer again for appreciating our work and for their valuable comments! If the reviewer has any further questions or follow-up thoughts, we are happy to engage in discussion.

---

### Official Review · Reviewer_WwUF · 2024-10-31

**Soundness:** 1
**Presentation:** 1
**Contribution:** 1
**Rating:** 3
**Confidence:** 4

**Summary:**

This paper identifies an explainability technique to better measure unlearning.

**Strengths:**

1. The notion of a concept vector as it relates to unlearning is novel to my knowledge.
2. The writing is clear, well formatted, and fluid.
3. The explanatory graphics are helpful and clear.

**Weaknesses:**

1. **Poorly posed problem** In section two, the authors try to define the problem, however there is very little concrete information about what type of unlearning this paper aims to address. It is clear to me it isn't exact unlearning, as that definition isn't in the paper and it's rather empirical. It doesn't seem to be the standard approximate unlearning either as the retrained model is not the point of comparison. This is made muddier by the idea that the goal isn't to remove the influence of specific data, but rather to remove a concept. To me this is far to vague to make scientific claims about.

2. **Evaluation Method is Hard to Parse** To the best of my ability to understand, the authors find (by hand) several subsets of the network's weights that posses the concepts we aim to forget. I think the evaluation is done by tracking the change in these parameters specifically. But if that's the evaluation, doesn't that present us with a method? Why can't we remove these weights? Set them to zero or perturb them in some other way?  I found this confusing throughout the paper. (I think this is Needle)

3. **Manual Nature** I see that the idea here is to create a benchmark so the fact that these concept vectors are found manually (which must have been a lot of work) doesn't present a major reproducibility problem, but it is odd and makes the whole benchmark quite subjective. Could it be that we are accidentally looking at unlearning the concepts easiest to isolate? Could that mean that the same methods might not work for other concepts?

**Questions:**

1. Can you define the unlearning problem a bit more clearly? Is the goal to end up with a model similar to one that was never trained on any data pertaining to the concept originally? If that is the goal, can you offer any way to gauge what data "pertainins" to a concept? If that's not the goal, what is?
2. Can you offer a way to find concept vectors for private individuals? This is someone mentioned only a few times in the training data who is likely to file a GDPR request. Would their concept vector be hard to find? Does it have to be found manually? Is there any guarantee that all people who could ask to be forgotten even have a concept vector? (This is critical as one of the strongest motivations for unlearning is this existing policy).
3. Can you explain why there isn't a circular argument in this paper? Does the needle method saturate the benchmark? If it does, what value is this benchmark to future methods and if it doesn't why is the benchmark useful if surgical manipulation of the concept vectors isn't the best approach according to this evaluation?
  -> In other words, is the optimal performer not that good?
4. Can you discuss how the order of methods differs here from other unlearning evaluation schemes? Is that difference reflect an intuitive problem with existing benchmarks?

---

> ### Author Response · Authors · 2024-11-21
> **Responses to the Reviewer WwUF (1/2)**
>
> We thank the reviewer for reviewing our work. We believe the reviewer misunderstood a few key points, including the overall idea of using concept vectors for evaluation and the proposed data generation pipeline. We attempt to clarify these points next and would be happy to answer any follow-up questions from the reviewer.
>
> > **W1 + Q1** (problem definition):
>
> We would like to firstly position our work within the broader literature in NLP. Indeed, the term “unlearning” conflates the formal notion the reviewer points to with the more empirically driven notion of this term in NLP. This paper follows a long series of work in NLP, which pointed out that factual knowledge (such as the identity of the capital of London) is stored in the fully connected layers in transformers. This observation sparked interest in “unlearning” this knowledge, i.e., making the model behave as if it no longer possesses it. This is similar to the problem of “unlearning” in general ML, but is naturally more empirical, as it is almost always not practical to retrain large LMs from scratch without being exposed to the concept.
>
> To be more concrete, our work follows the notion of unlearning or removal of from recent works about unlearning in LLMs [1, 2, 3, 4]. Given a concept c, which is typically a named entity, the goal is to remove any knowledge the model has about c. One common way to formalize “knowledge about c“ is through a knowledge-base like Wikidata, where facts are subject–relation-object triplets (e.g., the triplet <”Barack Obama”, year-of-birth, “1961”> would correspond to the fact “Barack Obama was born in Hawaii”). With this formalization, the task is essentially to remove any facts the model has where the subject is c. This setting has been used widely to formalize problems of knowledge editing and removal, and particularly was used for the MEMIT baselines employed in our work. While we agree that this formulation may be refined, this is not one of the contributions of our work, but rather this is the setting which has been studied recently and we largely build on top of it. Please note that our evaluation does compare the model responses/parameters before and after unlearning.
>
> > **W2 + Q3** (evaluation method):
>
> Needle is not an unlearning method, it is an oracle baseline. The reason it cannot be used as a general unlearning method is that it assumes the concept vector is given, while in practice it is not given and may not be trivial to find. Namely, finding the vector corresponding to a specific concept is challenging and generally an unsolved problem (due to superposition and features that are not aligned with standard bases, as we discuss in the paper’s limitations paragraph). However, it is often easy to identify vectors that encode some concepts, which is exactly what our paper leverages. If there’s some vector identified as capturing knowledge about Harry Potter – we want an unlearning method to remove this knowledge such that it is not recoverable anymore. Needle is applied as a baseline to demonstrate the importance of locating and removal of this residual knowledge, as we demonstrate in the context of jailbreak attacks.
>
> > **W3** (manual nature):
>
> The data selection process is mostly automated and systematic (which was appreciated by Reviewer 9s3u). There is indeed a manual selection phase, which aims to find the vectors that correspond to the most coherent concepts based on human judgment, followed by a causal validation step. While it does introduce a bias towards concepts that can be easily located (which we acknowledge in the limitations paragraph), our empirical results show that even in these cases unlearning methods perform badly in terms of removing parametric knowledge. This shows there is a large room for improvement of current unlearning methods. Regarding reproducibility – given that the benchmark relies on open-weight models, reproducing our pipeline is very easy. We also provide the specific indices of the located vectors.

---

> > ### Author Response · Authors · 2024-11-21
> > **Responses to the Reviewer WwUF (2/2)**
> >
> > **Regarding the reviewer's questions**:
> >
> > **Q1:** Please see our response to W1.
> >
> > **Q2:** Thanks for the question. If we already have data or queries related to a specific private individual, we can run the model on this data, collect the concept vectors with high activation during this process, and then validate them through causal experiments in model’s behaviors. This validation determines whether these concept vectors truly have specific causal relevance to the behavioral output of the target data and queries. Since data related to a private individual is often extremely sparse and rare, it must be in a superposition with other knowledge embedded in the same concept vector. Furthermore, the private individual may still be distributed across multiple concept vectors. However, in this paper, we focus on unlearning evaluation for one such concept vector, which we consider to represent the minimum scope of modification required for effective unlearning. For such data, it is extremely challenging to identify all the parameters in the model that encode this knowledge. However, we can at least pinpoint one of the most prominent concept vectors for evaluation.
> >
> > **Q3:** Please see our response to W2. In short – with the tools we have today, given a concept, it is hard to find a vector that encodes it (i.e. unlearning is hard), but given a vector it is often easy to inspect which concept it captures (i.e. evaluation is easy).
> >
> >
> > **Q4:**  For LLMs, nearly all current unlearning evaluations rely solely on the model's behavioral outputs. These evaluations typically require collecting queries that include the target knowledge and do not need to delve into the parameters' perspective. However, we argue that evaluation based entirely on the model’s behavior is often unreliable, as it cannot cover all possible relevant queries (e.g., jailbreak attacks). Moreover, the model might achieve the so-called unlearning effect merely by suppressing the retrieval or activation of the target knowledge (Table 4 in section 5.1), rather than truly erasing it, with the risk of this knowledge reactivating later on upon adversarial attacks. Therefore, we believe that evaluating from the root of knowledge storage—the parameters—is a reasonable complementary approach to assess whether the target knowledge has been thoroughly removed from the model.
> >
> > We hope our response clarifies our contributions, resolves the reviewer’s concerns, and that they will reconsider their evaluation. We are happy to engage in further discussion and answer any follow-up questions the reviewer may have.
> >
> >
> > ---
> > **References**:
> >
> > [1] Who's Harry Potter? Approximate Unlearning in LLMs.
> >
> > [2] Detecting pretraining data from large language models.
> >
> > [3] Guardrail baselines for unlearning in llms.
> >
> > [4] Large language model unlearning via embedding-corrupted prompts.

---

> > > ### Comment · Reviewer_WwUF · 2024-11-21
> > > **Reviewer Response**
> > >
> > > Thank you for the time and the thorough response. Unfortunately, I'm not convinced that there is a well posed problem here.
> > >
> > > If the idea is that ConceptVectors are the metric -- i.e. that a good unlearning method would match the effect on the parameters that the 'oracle' method Needle has -- then a compelling argument as to why this metric matches what we want from unlearning methods is missing.
> > >
> > > For example, the Needle outputs in the appendix are nonsense. Does the goal of unlearning in this work ignore model utility? Should we be satisfied with gibberish outputs? Additionally, if the goal is (in the authors words) "Given a concept c, which is typically a named entity, the goal is to remove any knowledge the model has about c." How well does a simple filter that scans the prompt and the generation for c and returns a fixed refusal prompt when c is present perform? This method seems to meet the criteria in this work even though it would have no effect on the model weights. In work aiming to build models that are no different from a retrained model, this method can be dismissed. Here however, your ConceptVectors would have precisely zero change, but your model output may never relate to c thus achieving the goal as stated.
> > >
> > >
> > > I think the problem statement is vague and the evaluations are not well motivated in this work.

---

> ### Author Response · Authors · 2024-11-21
> **Further Clarification for Reviewer WwUF (1/2)**
>
> We thank the reviewer for the prompt response and for engaging in discussion.
>
> **Regarding the statement: "a good unlearning method would match the effect on the parameters that the 'oracle' method Needle has"**
>
> Please note that this is inaccurate. Needle is an oracle in the sense that it has access to one (of potentially multiple) vectors capturing knowledge about the concept, but it may not necessarily introduce exactly the desirable effect. This is because (a)   there may be more vectors capturing knowledge about the concept, in which case Needle won't erase all the relevant knowledge, and (b) the way through which Needle modifies the concept vector (i.e., adding noise + training) may be artificial and incoherent with the model's internal representations.
>
> As the reviewer wrote -- the metric is based on changes in the ConceptVectors and specifically how their reflection through projection to the vocabulary. Needle is just one way to introduce changes to these vectors, which may not be the best one.
>
> Also in this sense, MEMIT serves as a strong explanation and a compelling argument, whose training style is more thoughtfully designed. It achieves significantly better results on our ConceptVectors metrics compared to other fine-tuning-based unlearning methods (Jaccard Similarity on LLaMA: 0.769 vs. 0.985) and also demonstrates superior unlearning performance in behavioral tests (Table 3 and Figure 3) and enhanced unlearning robustness against various adversarial attacks (Figure 9).
>
> **Regarding keeping model utility**
>
> - We agree that the model utility should be taken into account. This is part of the reason for including an evaluation of unrelated concepts in our benchmark, namely, we want to preserve the ability of the model to generally answer questions (i.e. generate text coherently and correctly) and the unrelated knowledge.
> - With respect to Needle:
>   - please note that it is just a rough baseline without sophisticated training: it operates only by adding random noise to disrupt the target concept vector and then applying simple fine-tuning adjustments. Also of note is that Needle will not hurt the model generating ability on the other unrelated knowledge questions.
>   - If necessary, we could design a new goal concept vector containing harmless knowledge to replace the original concept vector. This would allow for a reasonable adjustment and transformation of the knowledge stored in the original concept vector, enabling unlearning while avoiding nonsensical outputs when responding to the same knowledge-related questions. In this way, the model would output the carefully designed answers.
>   - However, this is not the goal of our paper. Our contribution is not to propose a perfect unlearning method but to highlight the direction that proper unlearning should take in the future. Specifically, it should involve making reasonable adjustments to the parameters where the knowledge is genuinely stored within the model. Only by doing so can more thorough unlearning be achieved, leading to stronger robustness against adversarial attacks.
>
> **Regarding the problem setup and the suggested filter approach**
>
> We are sorry to hear that the reviewer is not satisfied with this setup, especially since problems concerning editing and unlearning of concept knowledge have been widely studied in the research community in recent years -- example references are provided below.
> Currently, there are already various methods that can bypass such filter defenses.
> - The suggested filtering approach is not expected to work very well, since it is possible to extract information from the model without explicitly stating the concept name. For example, instead of feeding "Barack Obama", we could input the 44th president of the US.
> - Alternatively, the input sentence can be **multilingual**, in which case it may be difficult to design a filter that can filter all the low-resources languages. However, our experiments in Table 4 show proven that even a low-resource language will also apply high activations to the specific concept vector.
> - There are already many jailbreak methods capable of constructing nonsensical jailbreak prompts, often **consisting of random gibberish, scrambled characters, or disordered letters** [6, 7]. In such cases, simple filters struggle to detect these prompts effectively, yet they can still successfully induce the model to output the target knowledge. Our experiments have shown that, even in these scenarios, such prompts similarly enhance the activation of the concept vector corresponding to the target knowledge, making them identifiable at the parameter activation level (please see the experiment in W3 to 6fAd).
>
> Thank you once again for actively engaging in discussion with us.

---

> > ### Author Response · Authors · 2024-11-21
> > **Further Clarification for Reviewer WwUF (2/2)**
> >
> > ---
> > References:
> >
> > [1] Fast Model Editing at Scale. ICLR, 2022.
> >
> > [2] Locating and Editing Factual Associations in GPT. NeurIPS, 2022.
> >
> > [3] Mass-Editing Memory in a Transformer. ICLR, 2023.
> >
> > [4] Can Sensitive Information Be Deleted From LLMs? Objectives for Defending Against Extraction Attacks. ICLR, 2024.
> >
> > [5] Backward Lens: Projecting Language Model Gradients into the Vocabulary Space. EMNLP, 2024.
> >
> > [6] Universal and Transferable Adversarial Attacks on Aligned Language Models
> >
> > [7] AutoDAN: Generating Stealthy Jailbreak Prompts on Aligned Large Language Models. ICLR 2024

---

> > > ### Comment · Reviewer_WwUF · 2024-11-29
> > > **Final Thoughts**
> > >
> > > I think Reviewer 6fAd articulated the points I was trying to make well. Concept vectors can't be used to benchmark unlearning, the causal links are weak (see Reviewer 6fAd's comment for a better framing of these issues). Overall, this line of work is intersting, but there is a lot that needs to be done to take what is essentially an unverified interoperability method and use it for measurement like this. Even with the clarifications from the authors --- and I thank them for their time and effort --- my issues here have not be resolved and are larger than I'd expect could be resolved during a short rebuttal.

---

> ### Author Response · Authors · 2024-11-29
>
> We thank the reviewer for the prompt reply, for their patience, and for connecting their arguments to the points by Reviewer 6fAd, which we just responded to.
>
> > Concept vectors can't be used to benchmark unlearning, the causal links are weak
>
> We respectfully disagree. Needle introduces a causal effect to the model's predictions and a substantial decrease in the behavioral tests. We argue that a decrease of >0.5 in the BLEU score is significant, even if it's less prominent than other methods.
>
> > Overall, this line of work is intersting, but there is a lot that needs to be done to take what is essentially an unverified interoperability method and use it for measurement like this
>
> We appreciate that the reviewer finds our work interesting and agree that there is still a lot to be done -- we would like to emphasize that our work is, to the best of our knowledge, the first work to merge between interpretability methods and evaluation of unlearning and use parameter interpretation methods for this purpose. We see great potential in this direction and believe that our jailbreak experiments and analysis demonstrate its utility. As both unlearning and interpretability methods improve, the quality of benchmarks in this line will be improved as well. But one must start from somewhere and part of the message of this work is to argue that such intrinsic evaluations should be taken into account. We believe that the consistent trends and clear observations from our experiments provide evidence for the potential of this avenue.

---

### Official Review · Reviewer_9s3u · 2024-11-01

**Soundness:** 3
**Presentation:** 3
**Contribution:** 3
**Rating:** 6
**Confidence:** 4

**Summary:**

This paper introduces a new benchmark called ConceptVectors, for evaluating unlearning in large language models by tracking parametric knowledge traces. The key motivation for the work is that current unlearning evaluations rely primarily on behavioral tests, without considering whether the knowledge is actually erased from the model's parameters. The authors propose a methodology to identify "concept vectors" - specific parameter vectors that encode concrete concepts - and construct a benchmark containing hundreds of common concepts across two open-source LLMs. They found that existing unlearning methods mostly suppress concept knowledge during inference rather than truly removing it, making models susceptible to adversarial attacks that can recover the "unlearned" information. In contrast, directly ablating concept vectors proves more effective at genuine knowledge removal and improves robustness against jailbreaking attempts.

**Strengths:**

The paper builds an excellent automated pipeline to automatically discover directions in parameter space that encodes the knowledge about any concept.

Their evaluations are very thorough


The paper is very well presented

**Weaknesses:**

While the premise of concept vectors is very intriguing, the main qualitative evidence that concept vectors encode the knowledge of a concept in the models is not very clear. The numbers indicate that concept vectors are not really erasing the knowledge (high BLEU and Rouge scores).

It is slightly unclear what the concept vectors are really encoding? Is it the knowledge exclusively or something more higher level like "everything" about the concept including grammar and comprehension

**Questions:**

The text for Table 3 is very confusing. In section 4.3: "While gradient-based and preference-based optimization methods substantially restrict models from generating information about the concept (with Target QA and text completion scores < 0.25)" - so the BLEU scores <0.25 is considered that the model is behaviorally showing unlearning. But Needle has much higher BLEU scores but the authors claim that as unlearning too? ("while demonstrating prominent effect on the model’s outputs (50% − 70% decrease in QA performance)"). This is confusing. I could be missing something here - would love some clarity on this


The original space to search for concept vectors was very large as noted by authors and they propose some clever ways to filter this search space. Primarily "we sort the column vectors based on the average logit value in the projection to the vocabulary, Intuitively, this score indicates how strongly the vector promotes a specific concept". And they discard some non-important MLP layers to continue their search. What happens if you search in the discarded MLP space? It doesn't sound intuitive at all - but this could be a good experiment to show for strengthening the intuition claims.


The entire pipeline for finding concept vectors is very well designed and automated. I am curious as to how much time it takes for a new user to run this pipeline for erasing or analyzing a concept of their choice?

Previous unlearning methods like RMU (https://proceedings.mlr.press/v235/li24bc.html) show their efficiency in parameter space by training probes at each layer and showing that there is no knowledge traces. What are the authors views on this? Is this doing something similar to looking through lenses?


Some evaluation questions:
1. How does the Needle method work when prompted with multiple choice questions? For instance, if I remove Harry Potter and test MCQ accuracy on this dataset? How does the generation look for this dataset? Lynch et al found that prompting with context is a good attack method in some unlearning methods. This could be interesting to look at

2. Why is the unlearnt model (NEEDLE) outputting repeated patterns? (appendix Table 8.) Does this mean that concept vectors also encode grammar in the models? Not just the concept? It almost feels like NEEDLE is basically finding neurons responsible for concept generation and simply ablating it is also removing the "speaking ability". So maybe that says something about the model's mechanisms? I would like to know what the authors think about this?


I am majorly concerned about the Table. 3 results and would like more clarity on the evaluations what are the BLEU scores being calculated against. Happy to reconsider my rating after this clarity

---

> ### Author Response · Authors · 2024-11-21
> **Responses to the Reviewer 9s3u (1/3)**
>
> We thank the reviewer for their thorough and constructive review. We appreciate they found our data pipeline excellent, the experiments thorough, and the presentation compelling. We address the points and questions raised.
>
> > **W1:**
> - *“the main qualitative evidence that concept vectors encode the knowledge of a concept in the models is not very clear”* – our approach relies on several previous works that have established the existence of MLP vectors encoding specific concepts [1,2,3]. These studies systematically evaluate the presence of such vectors through vocabulary projections, and showed causal evidence of their contribution to the model’s predictions. Please also note that step 3 in our data pipeline is also meant to check exactly this – we causally validate that erasing the concept vectors influences the model’s ability to answer questions about the concepts but not on other concepts.
>
> - *“The numbers indicate that concept vectors are not really erasing the knowledge”* – We believe the reviewer may have misunderstood these results and we would like to explain. Our results (Table 3) show that Needle obtains mean target bleu scores of 0.46 and 0.31 for the two models. These are higher (worse) than the other unlearning methods (which get to 0.05-0.24). However, there are two important things to keep in mind:
>
>     - Needle only ablates **a single vector**, which constitutes **<0.001%** of the network parameters. Other methods are either not restricted and modify all the parameters (Finetuning Unlearning) or modify the MLP layers including the layer where the concept vectors are (MEMIT). For example, MEMIT modifies a total of **44K** MLP vectors (44K times more vectors than Needle), including the concept vector itself. Therefore, we consider the unlearning effect that Needle introduces to the model’s ability of generating concept knowledge is dramatic, showing that it does capture essential knowledge about the concept. As a reference, ablating a random vector in MLP layers keeps a BLEU score of **1.0**.
>
>     - The second important point is that we do not expect that the concept vector is the only parameters in the network that capture knowledge about the concept. Previous works show that knowledge is spread across multiple places in the network [3,4], so overall it is expected that ablating a single vector would not erase the knowledge completely. Needle is a baseline rather than a method for unlearning, and it is mostly to demonstrate the effect of directly ablating the parametric knowledge.
>
> It is important to note that, even though the concept vector is not necessarily the only place where a piece of knowledge is stored, it is still useful for evaluating unlearning – which is what we demonstrate in this work.
>
> > **W2:** *“It is slightly unclear what the concept vectors are really encoding?”*
>
> This is an interesting question but note it is orthogonal to the premise of the paper. Namely, as long as the concept vector captures unique knowledge about the concept, it should be removed by unlearning. Still, the fact that causally removing the concept vector generally prevents the model from answering questions about the concept may show that it captures general knowledge about it. Also, the concept vectors do not contain general grammatical or syntactic knowledge, as their removal does not affect the model's language modeling abilities and generating valid responses about unrelated concepts.
>
>
> **Regarding the questions raised:**
>
> **Q1:** *“The text for Table 3 is very confusing.”* – We apologize for the confusion! Please see our response above (the second point in our response to W1) for an explanation of why the decrease by Needle is non-trivial. Another thing to note is that BLEU and ROUGE are generally widely used scores for string matching, commonly used to compare a generated output with a reference string – 0 means the generated response is entirely different from the model’s original response, and 1 means it is exactly the same as before unlearning. So while a target QA score of 0.25 means a more dramatic decrease in the model’s ability to generate concept knowledge compared to a score of 0.5, a 0.5 score is also substantial as it means the responses of the model to questions about the concepts have substantially changed and often do not include the needed information.

---

> > ### Author Response · Authors · 2024-11-21
> > **Responses to the Reviewer 9s3u (2/3)**
> >
> > **Q2:** "*Searching in all MLP layers..*"
> >
> > Thanks for this question. Please note that we do not claim these layers are not important. We argue that they may capture more broad concepts which do not fall into our problem setting, such as syntactic and lexical features. We are interested in concepts that represent specific entities rather than, for example, language-related concepts which could (arguably) be much harder to remove from LLMs. Note that our selection also agrees with previous works [e.g., 5,6,7] that show that earlier layers capture more syntactic features and middle and upper layers capture more abstractive/semantic features. Also, when we apply projection in the filtered-out vectors from the middle-upper layers, we would see that their projection tokens are irregular patterns and do not revolve around the same concept. This could be due to the overlap of knowledge from multiple concepts, or it may stem from redundant parameters in the model, making them unsuitable for evaluating unlearning.
> >
> > **Q3:** “*How much time it takes for a new user to run this pipeline for erasing or analyzing a concept of their choice?*”
> >
> > Adding a new concept to our benchmark is a fast process when the related keywords for the concept are already available: (1) First, we simply load the model parameters and use vocabulary projection (this is done once) to quickly identify the corresponding concept vector candidates. This step is completed within **30 seconds** on average. (2) Next, we run the model using questions related and unrelated to the adding concept to perform causal validation on these vector candidates. This step takes around **60 seconds**. Totally, the whole process of takes within **90 seconds** on average. Deleting a concept from the benchmark is even faster, taking within **0.01** seconds as we have used the concept's name as the corresponding key for indexing.
> >
> > **Q4:** Regarding RMU – RMU, based on its loss function and underlying principles, does not aim to erase the parameters of target knowledge. Instead, it works by perturbing the model's activations on the target data, making it difficult for the model to process these knowledge and preventing its recall. This weakens the model’s ability to represent this knowledge in its internal hidden states, therefore making it harder for trained probes to extract the target knowledge from these disrupted hidden states. However, in this process, the actual storage of knowledge itself has not been disrupted, even though probing does not detect the target knowledge. We still see this as a potential risk.
> >
> > Our work approaches this problem by directly studying the parameters storing the target knowledge. By using the logits lens and causal validation, we identify parameters that store key knowledge and observe whether they are disrupted during unlearning. This allows us to assess to what extent the knowledge is erased from the model. We believe that starting from the knowledge storage provides a complementary thorough way to prevent the leakage of target knowledge, enabling a more complete unlearning process, rather than solely relying on the model’s behaviour.

---

> > > ### Author Response · Authors · 2024-11-21
> > > **Responses to the Reviewer 9s3u (3/3)**
> > >
> > > **Q5:** Regarding evaluation questions:
> > >
> > > - We expect that performance on multiple-choice questions (MCQ) will be similar to directly generating answers, as both require the model to recall and extract target knowledge. The difference is that in MCQ, the model needs to match the options to select the correct answer. To test this, we have conducted an additional experiment where we took 10 concepts and for each concept converted its corresponding 10 QA pairs into 10 multiple-choice questions. The original answers were retained as the correct options, while GPT-4 was used to generate three additional incorrect options for each question. We found that after using Needle to disrupt the storage of target knowledge, the model loses access to this knowledge and therefore fails to accurately match the correct answer in MCQ, instead randomly selecting an incorrect option. We will include this evaluation and examples in the appendix to improve the comprehensiveness of our analysis.
> > >
> > > - “*prompting with context is a good attack method in some unlearning methods*” – Please note that this in-context learning jailbreak attack is one of the methods evaluated in the paper (see Section 5). Results for this attack are consistent with the general trends, where attacks enhance the activation of the target concept vector (Table 4), thereby helping it retrieve the target knowledge.
> > >
> > > - “*Why is the unlearnt model (NEEDLE) outputting repeated patterns?*” – Thanks for the great question. Needle will add noise in a random direction to the target concept vector, which alters the embedded knowledge within the vector into something unknown and potentially irregular. Moreover, during this noise injection process, the model’s high activation toward this concept vector remains unchanged. As a result, the model continues to rely on this disrupted vector for generating responses. The newly embedded knowledge within the altered vector then dominates the outputs, which lead to the occurrence of repetitive patterns. Notably, ablating the concept vectors do not affect the model’s generation of answers to questions about unrelated knowledge, **demonstrating that the knowledge it contains is specific rather than general**.
> > >
> > > We thank the reviewer again for the valuable questions.
> > >
> > > ---
> > > **References:**
> > >
> > > [1] Transformer Feed-Forward Layers Build Predictions by Promoting Concepts in the Vocabulary Space. EMNLP 2022.
> > >
> > > [2] LM-Debugger: An Interactive Tool for Inspection and Intervention in Transformer-Based Language Models. EMNLP 2022.
> > >
> > > [3] Knowledge Neurons in Pretrained Transformers. ACL 2022.
> > >
> > > [4] Dissecting Recall of Factual Associations in Auto-Regressive Language Models. EMNLP 2023.
> > >
> > > [5] BERT Rediscovers the Classical NLP Pipeline. EMNLP 2019
> > >
> > > [6] Transformer Feed-Forward Layers Are Key-Value Memories. EMNLP 2021.
> > >
> > > [7] The Semantic Hub Hypothesis: Language Models Share Semantic Representations Across Languages and Modalities. 2024.

---

> ### Comment · Reviewer_9s3u · 2024-11-23
>
> I appreciate the authors responding to all the questions! Especially, Table 3 makes a lot of sense to me now. Thanks for the clarity!
>
> The response by authors has made it clear that the metric used is measuring the text similarity between edited and original model's generation. While I do agree with author's view that: "truly unlearnt model should not exhibit concept vector" - I think there is also a more important point that Table 3 could make. "is concept vector truly representing the concept?". If concept vector truly captured the concept - I would expect that ablating concept vector would naturally reduce QA performance on the related subject but not on unrelated. But BLEU scores validate only text similarity - so it makes it slightly unclear if the metric is validating the correctness of the answer or the change in the answering style.
>
> How do the authors argue that BLEU is precisely measuring the correctness of the answers?
>
>
> EDIT: I missed the response from authors where they measure the MCQ accuracy and found that the answers are random when ablating NEEDLE. I very much appreciate this experiment! and I believe it answers the question of "correctness". But the sample size of 10 might not be enough to reach to a stronger conclusion. I do believe it is important to get this abstraction right and show that concept vectors truly capture the semantics of the concept.

---

> > ### Author Response · Authors · 2024-11-23
> > **Further Clarification for Reviewer 9s3u**
> >
> > Thank Reviewer 9s3u for actively engaging in our discussion and provding the valuable feedback!
> >
> > > *“If concept vector truly captured the concept - I would expect that ablating concept vector would naturally reduce QA performance on the related subject but not on unrelated.”*
> >
> > We agree that if a concept vector truly represents at least a part of the knowledge associated with the corresponding concept, then ablating the concept vector should exhibit specificity: the model's performance on related knowledge questions should degrade significantly more than on unrelated questions. This conclusion is demonstrated and validated in Figure 4 and Section 4 in our paper.
> >
> > > *“But BLEU scores validate only text similarity - so it makes it slightly unclear if the metric is validating the correctness of the answer or the change in the answering style. How do the authors argue that BLEU is precisely measuring the correctness of the answers?“*
> >
> > - The primary goal of this evaluation is not factuality but unlearning, namely, we want that the answers the model produces are different from those it produced before unlearning. An important note in this regards is that the answers for this evaluation typically include the **named entities** (e.g., Q: Who is the author of Harry Potter books? A1: **J.K. Rowling**. A2: It is **J.K. Rowling**. A3: **J.K. Rowling** is the author). Thus, when we observe very low BLEU scores, it suggests that the unlearned model's answers likely no longer contain the original correct named entities. The lower the BLEU score, the higher the probability that the correct named entities are absent in the unlearned model's answers. Even though it is an approximation, this metric is automated and efficient, providing a numeric and continuous measure of how the model's responses change during the unlearning process.
> >
> > - Another point for consideration is that because the questions in ConceptVectors are simple and about commonly known facts, about 94% of the models’ responses (based on our qualitative evaluation) are also factually correct and align with the ground truths, which means that the evaluation in practice also verifies the answers’ correctness. This follows the behavioral evaluations conducted in prior works [1, 2, 3].
> >
> > Thanks again for your comments, we are happy to address any other questions the reviewer may have. Given the new results and our response, we would appreciate if they reconsider their evaluation and increasing their score.
> >
> > ---
> > **References:**
> >
> > [1] Negative Preference Optimization: From Catastrophic Collapse to Effective Unlearning. COLM 2024
> >
> > [2] SOUL: Unlocking the Power of Second-Order Optimization for LLM Unlearning. ACL 2024.
> >
> > [3] Large Language Model Unlearning. NeurIPS 2024.

---

> > > ### Comment · Reviewer_9s3u · 2024-11-25
> > >
> > > Thanks for pointing me to Figure 4! That is what I expected with ConceptVectors if they were to represent the concept truly. So why are the results different in Table . 3? If I am reading it right, Table.3 shows NEEDLE which also does gradient ascent on top of vector noising. But figure.4 doesn't do gradient ascent?
> > >
> > > NEEDLE seems to be reducing both the target and untargeted content.
> > >
> > > Regarding - "The primary goal of this evaluation is not factuality but unlearning, namely, we want that the answers the model produces are different from those it produced before unlearning" - I think the authors would agree that an unlearnt model when asked about erased concept, would no longer generate the correct answer. The only concern is that the model could be generating the answer with either rewording or paraphrasing. I can understand the motivation behind BLEU given the answers are mostly factual questions with obvious answers. However, I would urge the authors to consider the MCQ experiment post rebuttal, if time permits. But for the rebuttal phase, I do not recommend any further experiments.

---

> ### Author Response · Authors · 2024-11-23
> **Updation on RMU experiments for Reviewer 9s3u**
>
> Dear Reviewer 9s3u,
>
> Thanks again for mentioning the RMU method [1] in Q4 in your initial review. This message is to update the reviewer that we have conducted additional experiments, evaluating RMU (representation engineering-type method) as part of our main results. Below are the experiment settings and results, which we will make sure to incorporate to the paper in it's next revision.
>
> **Experiment Settings:**
>
> We evaluated RMU on the ConceptVectors benchmark. The results on the LLaMA2-7B-chat portion are provided in the table below.
>
> There are two points to note:
>
> - The original RMU method will edit the **same fixed layers for all the samples** (the default hyperparameters are the 5-th, 6-th, 7-th layers in a 32 layers model). So we first report its performance using these original settings.
> - To achieve better unlearning effects on the corresponding concept vectors, we also modified the default hyperparameters of RMU, where for each concept we edit the layer containing its concept vector, as well as the two preceding layers. We named this alternative RMU as **RMU_enhanced** in the following table.
>
> In the table, we further include two representative methods described in our paper, DPO and MEMIT+Empty, for comparison.
>
> **Results:**
>
> | LLaMA2-7B-chat | Jaccard Similarity | BLEU | Rouge-L | Unrelated BLEU | Unrelated Rouge-L |
> |----------------------|--------------------|------|---------|-----------------|-------------------|
> | DPO         |  0.983              | 0.179    | 0.377    |0.263       | 0.461      |
> | MEMIT+Empty          | **0.725**              | **0.087**|  **0.207**   | **0.379**           | **0.565**             |
> | RMU                  | 0.999              | 0.157| 0.410   | 0.204           | 0.459             |
> | RMU_enhanced        | **0.722**              | 0.129| 0.269   | 0.253           | 0.487             |
>
>
> From the results we make the following observations:
>
> 1. Overall, in terms of behavioural evaluation, we observe that MEMIT continues to deliver the best performance, achieving the most effective unlearning (lowest BLEU and ROUGE scores) while also preserving unrelated knowledge well in the model (highest unrelated BLEU and ROUGE scores).
> 2. In RMU’s original setting, the fixed layers prevent unlearning from effectively influencing concept vectors in other layers. As a result, it performs poorly in terms of our intrinsic evaluation (Jaccard Similarity: 0.999). In terms of behavioural evaluation, its performance is comparable to DPO but outperformed by MEMIT and RMU_enhanced.
> 3. In RMU_enhanced, we dynamically select the layers for editing for each concept, therefore it has a better intrinsic performance (lower Jaccard Similarity), indicating a more effective influence on the concept vectors. On the behavioral evaluation, the enhanced RMU achieves improved unlearning performance (lower BLEU and ROUGE scores) and better preservation of unrelated knowledge (higher BLEU and ROUGE scores) compared to the original RMU configuration. These results demonstrate that targeting the specific parameters where knowledge is actually stored in the model allows for superior unlearning effects while minimizing the impact on unrelated knowledge.
>
> Additionally, it is worth noting that while RMU_enhanced and MEMIT+Empty show similar performance on the Jaccard Similarity metric, suggesting a comparable degree of parameter modification, MEMIT+Empty achieves better unlearning results on behavior metrics. This can be attributed to MEMIT’s potentially more effective loss function, which introduces more optimal directional modifications and edits to the concept vectors.
>
> We will incorporate these additional results in a revised version of the paper.
>
> Please feel free to raise any points that are unclear to you. We sincerely appreciate your engagement in the discussion and are happy to know if our response and new results change anything about your assessment of our work.
>
> ---
> **References:**
>
> [1] The wmdp benchmark: Measuring and reducing malicious use with unlearning. ICML 2024

---

> ### Author Response · Authors · 2024-11-27
> **Further Clarification for Reviewer 9s3u about the Needle performance and the rewording problem**
>
> Thank you so much for actively participating in our discussion and continuing to provide valuable feedback!
>
> > *Table.3 shows NEEDLE which also does gradient ascent on top of vector noising. But figure.4 doesn't do gradient ascent? NEEDLE seems to be reducing both the target and untargeted content.*
>
> Thank you for raising this important point.
>
> **About the Experiment Settings:**
>
> Yes, in the Figure 4 experiment, we conducted causal validation experiments to verify the causal relationship between the concept vector and its corresponding concept. Therefore, the experimental setup here involved **adding noise solely** to the target concept vector.
>
> In Table 3 experiment, the Needle approach will apply gradient ascent to fine-tune the target concept vector after the noise adding, while keeping all other model parameters unchanged.
>
> **About the Needle performance:**
>
> Thank you for noticing this.
>
> To make it easier to compare, we have added the results from the setting that only involves disrupting the concept vector with noise (Figure 4) into Table 3, as destorated below:
>
> | LLaMA2-7B-chat | Target QA (BLEU / Rouge-L) | Unrelated QA (BLEU / Rouge-L) |
> | --- | --- | --- |
> | Gradient Ascent | 0.119 / 0.347 | 0.169 / 0.377 |
> | MEMIT (Empty response) | 0.087 / 0.207 | 0.379 / 0.565 |
> | **Adding Noise Alone** | 0.532 / 0.672 | **0.947** / **0.973** |
> | Needle | 0.462 / 0.588 | 0.534 / 0.678 |
> | **OLMo-7B** | - | - |
> | Gradient Ascent | 0.056 / 0.538 | 0.057 / 0.549 |
> | MEMIT (Empty response) | 0.121 / 0.253 | 0.316 / 0.471 |
> | **Adding Noise Alone** | 0.331 / 0.553 | **0.786** / **0.887** |
> | Needle | 0.313 / 0.726 | 0.447 / 0.689 |
>
> From the table above we can observe that:
>
> - Compared to the "Adding Noise Alone" approach, the Needle method can achieve better unlearning performance (lower Target QA BLEU and ROUGE-L scores), but it also indeed results in a greater loss of performance on unrelated knowledge (lower Unrelated QA BLEU and ROUGE-L scores).
>
> - The underlying reason for this is likely: due to the limitation of gradient ascent, where the gradient optimization direction is relatively random and uncontrolled. As a result, it may inadvertently encode unrelated content into the concept vector, which impacts performance on other tasks and leads to a greater effect on the unrelated scores.
>
> - From the table, we can infer again that parameter-based unlearning methods have enormous future potential: the "Adding Noise Alone" approach, compared to all other baselines, **creates the greatest difference between target knowledge** (Target QA BLEU and ROUGE-L scores) **and the model's unrelated knowledge** (Unrelated QA BLEU and ROUGE-L scores) (LLaMA BLEU and Rouge-L: **0.415 | 0.301**; OLMo BLEU and ROUGE-L: **0.334 | 0.455**). This suggests that it achieves the **best trade-off** between preserving unrelated knowledge and erasing target knowledge.
>
> - This result further strengthens our claim that performing unlearning on target knowledge parameters is the key direction for achieving more thorough unlearning in the future.
>
> Here, we emphasize again that Needle is merely a **baseline** method we proposed—a preliminary approach rather than a fully optimized solution. It has significant room for improvement. In this paper, we will remove the gradient ascent training step in the Needle method to restore its inherent strong trade-off between preserving unrelated knowledge and removing target knowledge, and update the corresponding section on Needle's performance. In future work, we plan to design more refined steps to further enhance the Needle method and improve its performance.
>
> > *The only concern is that the model could be generating the answer with either rewording or paraphrasing… I would urge the authors to consider the MCQ experiment post rebuttal, if time permits.*
>
> Thanks again for your suggestion about the Multi-choise Questions experiments.
>
> We believe your concerns regarding the rephrasing or rewording of questions can be addressed and explained **to some extent** by the experiments on our low-resource language jailbreak attacks presented in **Sections 5.1 and 5.2**. In these experiments, we used another language (German) to query the LLM about the same knowledge, focusing on the same concept vectors. We observed that these concept vectors exhibited high activation even when questions were posed in German (Section 5.1). Moreover, ablating these concept vectors resulted in effective unlearning for the German queries as well (Section 5.2).
>
> Therefore, we can reasonably infer that the unlearning process is effective regardless of how a given question is rephrased, or even when posed in a **different language**, as long as it pertains to the same underlying knowledge. Ablating the corresponding knowledge parameters appears to consistently impact the associated knowledge.
>
> That said, we appreciate your suggestions and will include the MCQ experiment in the revised version of paper to make our work more persuasive.

---

### Official Review · Reviewer_1G7R · 2024-11-02

**Soundness:** 2
**Presentation:** 3
**Contribution:** 2
**Rating:** 5
**Confidence:** 3

**Summary:**

The paper introduces a dataset of vectors associated with different concepts, which can be used to test whether an LLM has a knowledge about a topic. This is particularly relevant for unlearning, where these vectors can be considered an audit tool to check whether the knowledge has been truly removed or just suppressed.

The vectors are obtained by selecting some columns in the output of some MLP layer, based on the average logit score of the columns (after multiplying by the output embedding). The scores are checked with GPT-4 and manually.

Concept vectors are then used to evaluate how effective some common unlearning methods are, and it is proposed a new method, Needle, that perturbs the concept vector associated with the concept to be unlearned. The conclusion is that current unlearning methods obfuscate the knowledge rather than removing it. Needle turns out to be robust to adversarial attack.

**Strengths:**

Concept vectors might be useful to locate knowledge associated with a specific topic and used as an audit tool to check whether unlearning happened successfully. More generally, locating specific knowledge might help in interpreting the behavior of LLMs.

**Weaknesses:**

- The score description should be more precise. For example it’s not super clear how to judge whether a single token is relevant to a topic. A single token can be relevant to multiple topics.
- It’s not clear how the vectors are selected. After the score is computed, are the vectors selected for a fixed layer? Does the layer depend on the concept?
- The Needle unlearning baseline is similar to RepNoise and RMU, so an additional comparison with these two methods would be useful. They seem to be robust to jailbreak attacks too.
- It’d be good to include an evaluation of suffix based adversarial attacks, like GCG
- It’d be good to find concept vectors for a bigger set of models

**Questions:**

- There exist methods that try to locate concepts by looking at the internal activations (activation addition, representation engineering etc). How does your method compare to those?

---

> ### Author Response · Authors · 2024-11-21
> **Responses to the Reviewer 1G7R (1/2)**
>
> We thank the reviewer for their feedback and are encouraged by their appreciation of concept vectors being useful for auditing unlearning.
>
> **Regarding the mentioned weaknesses:**
>
> > **W1:** "*The score description should be more precise...A single token can be relevant to multiple topics.*"
>
> We assume the reviewer refers to the token scores – this score is essentially the logit corresponding to a certain token after the projection. Namely, given an value vector v of dimension d, after projecting it to the vocabulary space (E * v) we get a vector r of dimension |V|, where V is the vocabulary. The vector r has an entry for each token with its corresponding logit. The logits are the scores – we sort them by value and consider the top-scoring tokens which often show a clear pattern that described a concrete concept. This observation has been reported in previous works [1,2]. Please note that the topics are inferred from the whole set of top-scoring tokens, rather than single tokens. While a single token may be associated with multiple topics, we can identify the most relevant concept by analyzing the entire set of projection tokens, and validate it using causal experiments.
>
> > **W2:** "*It’s not clear how the vectors are selected...Does the layer depend on the concept?*"
>
> Thanks for these questions. When selecting the concept vectors candidates, we do not focus on which layer they are locate in. Instead, we are primarily concerned with whether a subset of their projection tokens is highly relevant to a specific theme. Subsequently, we validate their causal relationship and specificity regarding the corresponding knowledge output through causal validation experiments in behaviors. Therefore, the layer in which a concept vector is located not a major factor in its selection. That said, in practice, we have observed that vectors for specific concepts with a substantial causal effect are more commonly found in the middle-upper layers of the model. The selection process is described in Section 3, and we are happy to provide more details if needed.
>
> > **W3:** "*The Needle unlearning baseline is similar to RepNoise and RMU...They seem to be robust to jailbreak attacks too.*"
>
> We thank the reviewer for this valuable suggestion. We have been working on these additional experiments following this suggestion, and aim to share the results soon. In this context, please see our response in Q4 to Reviewr 9s3u where we further discuss how our work related to representation engineering-based methods.
>
> > **W4:** "*It’d be good to include an evaluation of suffix based adversarial attacks, like GCG*"
>
> Thanks for this comment! We have followed the reviewer’s suggestion and provide the effects of two current representative Jailbreak methods – GCG [4] and AutoDAN [5] – on the activation enhancement of concept vectors, and compare them with the effects without jailbreak and with the first crafted prompt in the paper. We observe similar trends, where jailbreak enhances the activation of model knowledge parameters.
> We will add these results to the paper.
>
> | Model / Attack | No Jailbreak | Crafted Prompt1 | GCG | AutoDAN |
> |-------|-------|-------|-------|-------|
> | Unlearned via Gradient Difference | 2.14 | 3.07 | 3.51 | 3.20 |
> | Unlearned via DPO | 1.42 | 2.03 | 2.92 | 2.65 |
> | Vanilla | 2.59 | 3.34 | 4.02 | 3.84 |
>
> > **W5:** "*It’d be good to find concept vectors for a bigger set of models*"
>
> Please note that the purpose of the benchmark is to evaluate the performance of different unlearning methods, rather than to evaluate model performance. Thus, we focus on covering a diverse set of unlearning methods rather than models, including 9 different baselines from 4 different categories. In addition, the overall approach of using concept vectors has been proven to be generally applicable. Their existence has been shown in multiple transformer-based language models, including GPT2 [1,2,3] and GPT-J [3], and in the appendix (see Table 7) in our paper, **where we have validated that concept vectors can be located in ChatGLM3-6B, Mistral-7B, Qwen1.5-7B and even Qwen1.5-72B**. Therefore, it should be easy for future work to extend this benchmark in case they are interested in unlearning for specific models.

---

> > ### Author Response · Authors · 2024-11-21
> > **Responses to the Reviewer 1G7R (2/2)**
> >
> > **Regarding the reviewer’s question:**
> >
> > **Q1:** Thanks for this question. The methods mentioned (activation addition and representation engineering) are employed for steering the model’s behavior or locating features in the representation space. It is worth noting that the main purpose of representation engineering methods (such as RMU) and activation addition methods in the context of unlearning is not to directly erase the knowledge stored in the model's parameters. Instead, their primary goal is to perturb the model's activations on the target data, making it harder for the model to process and recall this knowledge. By weakening the model's ability to represent this knowledge in its internal hidden states, these methods achieve a defensive effect. In contrast, our work focuses on evaluating directly whether the knowledge stored in the model's parameters—the source of the knowledge—has been effectively affected during unlearning. We argue that this approach represents a valuable path to explore toward achieving more robust unlearning, reducing associated risks.
> >
> > ---
> > **References:**
> >
> > [1] Transformer Feed-Forward Layers Build Predictions by Promoting Concepts in the Vocabulary Space. EMNLP 2022.
> >
> > [2] Analyzing Transformers in Embedding Space. ACL 2023.
> >
> > [3] Dissecting Recall of Factual Associations in Auto-Regressive Language Models. EMNLP 2023.
> >
> > [4] Universal and Transferable Adversarial Attacks on Aligned Language Models
> >
> > [5] AutoDAN: Generating Stealthy Jailbreak Prompts on Aligned Large Language Models. ICLR 2024

---

> > > ### Comment · Reviewer_1G7R · 2024-11-22
> > > **Follow up on Q1**
> > >
> > > **Q1**: thank you for making the distinction more clear. I'd suggest to add just a couple of lines in the paper highlighting this difference.

---

> > > > ### Author Response · Authors · 2024-11-23
> > > > **Further Clarification for Reviwer 1G7R (2/2)**
> > > >
> > > > **Q1:**
> > > >
> > > > > I'd suggest to add just a couple of lines in the paper highlighting this difference.
> > > >
> > > > Thanks for the suggestion. We will add the following explanation in Section 4:
> > > >
> > > >     Additionally, there are other methods based on representation engineering [1, 2, 3] and activation modification [4] that aim to unlearn or refuse generating harmful knowledge by adjusting the internal representations of the model. Notably, the primary goal of these methods is to perturb the model's activations on the target data, making it more difficult for the model to process and recall the knowledge, rather than erasing it from the model's parameters. By weakening the model's ability to represent this knowledge within its internal hidden representations, these methods achieve a defensive effect. In contrast, our work focuses on evaluating whether the knowledge stored in the model's parameters has been effectively impacted by existing unlearning methods.
> > > >
> > > > As we addressed the reviewer's concerns, we would sincerely appreciate it if they can reconsider their evaluation and increasing their score.
> > > >
> > > > ---
> > > > **References:**
> > > >
> > > > [1] The wmdp benchmark: Measuring and reducing malicious use with unlearning. ICML 2024
> > > >
> > > > [2] Improving alignment and robustness with circuit breakers.
> > > >
> > > > [3] Refusal in language models is mediated by a single direction. NeurIPS 2024
> > > >
> > > > [4] Representation noising: A defence mechanism against harmful finetuning. NeurIPS 2024

---

> > > > > ### Author Response · Authors · 2024-12-04
> > > > > **Updation on RepNoise Experiment and Kind Reminder for considering our Revisions**
> > > > >
> > > > > Thank again for Review 1G7R's constructive suggestions, especially about the Representation Engineering Unlearning baselines.
> > > > >
> > > > > Here, in addition to completing the RMU [1] experiments and incorporating them into the main body of the paper, we have also **newly completed the RepNoise [2] baseline experiment suggested by Reviewer 1G7R**. We evaluated the RepNoise method on our ConceptVectors benchmark, and the results on the llama2-7B-chat model are shown in the table below:
> > > > >
> > > > > | LLaMA2-7B-chat | Jaccard Similarity | BLEU | Rouge-L | Unrelated BLEU | Unrelated Rouge-L |
> > > > > |----------------------|--------------------|------|---------|-----------------|-------------------|
> > > > > | DPO         |  0.983              | 0.179    | 0.377    |0.263       | 0.461      |
> > > > > | MEMIT+Empty          | **0.725**              | **0.087**|  **0.207**   | **0.379**           | **0.565**             |
> > > > > | RMU                  | 0.999              | 0.157| 0.410   | 0.204           | 0.459             |
> > > > > | RMU_enhanced        | **0.722**              | 0.129| 0.269   | 0.253           | 0.487             |
> > > > > | RepNoise | 0.956         | 0.140 | 0.297  | 0.269           | 0.468             |
> > > > >
> > > > > Analysis of the results is as follows:
> > > > >
> > > > > - First, it is important to note that, like RMU, RepNoise also aims to remove information about harmful tasks from intermediate activations, making it difficult for the model to use this harmful information. The goal of it is not to directly erase target knowledge from the model's parameters.
> > > > > - RepNoise targets on the MLP activations at each layer, and during training, it does not restrict updates solely to the MLPs but instead on the full parameters of the models. As a result, we observe that its modification effect on concept vectors is relatively small (Jaccard: 0.956), confirming that it does not directly erase knowledge stored in the parameters.
> > > > > - Although RepNoise also involves full parameter fine-tuning, its objective function is designed more precisely compared to DPO. This design reduces the impact on irrelevant knowledge while enhancing the unlearning effects on target knowledge. However, due to its insufficient modification effect on the target knowledge parameters, its performance is not as effective as Needle, MEMIT, and RMU_enhanced.
> > > > >
> > > > > **Given that we have fully addressed the reviewer's concerns (**W1, W2**), completed all suggested experiments (**W3, W4, W5**), and incorporated the recommended analyses into the updated paper (**Q1, W3, W4, W5**), we would greatly appreciate it if the reviewer could consider our improvements and reconsider their score.**
> > > > >
> > > > >
> > > > > ---
> > > > > **References**
> > > > >
> > > > > [1] The wmdp benchmark: Measuring and reducing malicious use with unlearning. ICML 2024
> > > > >
> > > > > [2] Representation Noising: A Defence Mechanism Against Harmful Finetuning. NeurIPS 2024

---

> > ### Comment · Reviewer_1G7R · 2024-11-22
> > **Follow up on W1+W2**
> >
> > Thank you for following up and for providing detailed responses.
> >
> > - **W1 + W2**: I understand how the score is computed, however the score's formula $\sum_{i=1}^{\mathcal{V}}(e_i \cdot v_j^{\ell}) / \mathcal{V}$ seems to be computed at each layer $\ell$. If I understand correctly, the formula seems to compute the token score as if the model was “chopped” at the $\ell$th layer. That's why it looks to me that the score depends on the layer. Maybe I’m missing something here.
> >
> > - **W3 + W4**: thank you for considering adding the RMU and RepNoise evaluations. Also, great to see that GCG and AutoDAN attacks also have a similar effect on the activations.

---

> ### Author Response · Authors · 2024-11-23
> **Further Clarification and Updated Experiment on RMU for Reviwer 1G7R (1/2)**
>
> Thanks Reviewer 1G7R for the timely reply and follow up questions, which we happy to clarify.
>
> **Concept vectors’ scores (W1+W2):**
>
> > *“however the score's formula  ∑i=1V(ei⋅vjℓ)/V seems to be computed at each layer ℓ. If I understand correctly, the formula seems to compute the token score as if the model was “chopped” at the ℓ-th layer. That's why it looks to me that the score depends on the layer.”*
>
> Our selection process strives to identify the vectors that exhibit the most prominent and specific concepts across the network, which we approximate based on the logits in the vocabulary projection. However, the logits of vectors in different layers are not comparable to each other, because they reside in different representation spaces. This is why we first take the most “promising” vectors per layer and then from the remaining vectors across the whole network we select those where we see the clearest concepts based on GPT, while disregarding the layer information.
>
> **RMU evaluations (W3+W4):**
>
> We evaluated RMU on the ConceptVectors benchmark according to the reviewer’s suggestion. The results on the LLaMA2-7B-chat portion are provided in the table below.
>
> There are two points to note:
>
> - The original RMU method will edit the **same fixed layers for all the samples** (the default hyperparameters are the 5-th, 6-th, 7-th layers in a 32 layers model). So we first report its performance using these original settings.
> - To achieve better unlearning effects on the corresponding concept vectors, we also modified the default hyperparameters of RMU, where for each concept we edit the layer containing its concept vector, as well as the two preceding layers. We named this alternative RMU as **RMU_enhanced** in the following table.
>
> In the table, we further include two representative methods described in our paper, DPO and MEMIT+Empty, for better comparison. From the results we make the following observations:
>
> 1. Overall, in terms of behavioural evaluation, we observe that MEMIT continues to deliver the best performance, achieving the most effective unlearning (lowest BLEU and ROUGE scores) while also preserving unrelated knowledge well in the model (highest unrelated BLEU and ROUGE scores).
>
> 2. In RMU’s original setting, the fixed layers prevent unlearning from effectively influencing concept vectors in other layers. As a result, it performs poorly in terms of our intrinsic evaluation (Jaccard Similarity: 0.999). In terms of behavioural evaluation, its performance is comparable to DPO but outperformed by MEMIT and RMU_enhanced.
>
> 3. In RMU_enhanced, we dynamically select the layers for editing for each concept, therefore it has a better intrinsic performance (lower Jaccard Similarity), indicating a more effective influence on the concept vectors. On the behavioral evaluation, the enhanced RMU achieves improved unlearning performance (lower BLEU and ROUGE scores) and better preservation of unrelated knowledge (higher BLEU and ROUGE scores) compared to the original RMU configuration. These results demonstrate that targeting the specific parameters where knowledge is actually stored in the model allows for superior unlearning effects while minimizing the impact on unrelated knowledge.
>
> Additionally, it is worth noting that while RMU_enhanced and MEMIT+Empty show similar performance on the Jaccard Similarity metric, suggesting a comparable degree of parameter modification, MEMIT+Empty achieves better unlearning results on behavior metrics. This can be attributed to MEMIT’s potentially more effective loss function, which introduces more optimal directional modifications and edits to the concept vectors.
>
> | LLaMA2-7B-chat | Jaccard Similarity | BLEU | Rouge-L | Unrelated BLEU | Unrelated Rouge-L |
> |----------------------|--------------------|------|---------|-----------------|-------------------|
> | DPO         |  0.983              | 0.179    | 0.377    |0.263       | 0.461      |
> | MEMIT+Empty          | **0.725**              | **0.087**|  **0.207**   | **0.379**           | **0.565**             |
> | RMU                  | 0.999              | 0.157| 0.410   | 0.204           | 0.459             |
> | RMU_enhanced        | **0.722**              | 0.129| 0.269   | 0.253           | 0.487             |
>
> We will incorporate these additional results in a revised version of the paper.

---

### Official Review · Reviewer_hnu6 · 2024-11-03

**Soundness:** 2
**Presentation:** 3
**Contribution:** 3
**Rating:** 6
**Confidence:** 4

**Summary:**

The submission presents a new benchmark for Machine Unlearning, with a specific focus on ensuring data is removed from the model weights, not just from the model's behavior. The benchmark finds concepts represented in model weights, synthetically generates questions about that concept and retrieves Wikipedia articles about the concept, then validates the concept vectors ablating them and observing the impacts on the collected data. Concept vectors which have large but targeted impact on the concept are taken as unlearning targets. The benchmark then evaluates unlearning algorithms based on whether they remove the concept vector (based on change in predicted tokens), as well as standard behavioral tests.

**Strengths:**

1.	Submission targets an important and relevant question: do unlearning algorithms actually remove the information they claim to?
2.	Submission is well-written and results are presented well
3.	Provides interesting analysis on the effect of jailbreak attacks on model internals post-unlearning.
4.	It was interesting to use MEMIT as an unlearning method to remove internal concept representations.

**Weaknesses:**

1.	The authors claim that "This work presents the first benchmark for internal evaluation of unlearning methods." However, there is at least one benchmark [1] that trains linear probes on the activations of the models, trying to predict answers to held-out questions based on the model's internal representations. This submission cites [1] in Section 6 but does not evaluate against the method from [1] that address some of the concerns raised about other unlearning methods not removing internal concept representations.
2.	Submission uses the MLP basis for selecting concept vectors, which does not seem ideal for finding concept vectors, as it restricts concept vectors to the neuron basis and there is a consistent view in the mechanistic interpretability literature that neurons are polysemantic. If the concept vectors are polysemantic, then it is possible that unlearning algorithms may avoid removing them to prevent losses in other tasks
3.	There are a few of components in the dataset generation process that required manual input that seemed a bit vague in definition:
a) In 3.1 Step 1: “Last, we (authors) manually review the top-scoring vectors and select those exhibiting a clear pattern corresponding to a concrete and specific concept.” – what is the definition of a “clear pattern” and a “concrete and specific concept”?
b) In 3.3: “we manually verify that the questions are about the given concept and that they are simple and reasonable.”
4.	Specific tokens may not be the best way of describing a concept because with individual tokens it can sometimes be ambiguous how they are used contextually.
5.	It is unclear what quality-control measures when generating the behavioral tests. I think this stage, where concepts are generalized from single tokens to entire questions or paragraphs, is a place where things could go wrong.
6.	At a higher level, it is unclear how deriving concepts directly from the model causes differences from pre-selecting concepts. It seems plausible that taking concepts directly from model parameters creates concepts that don't align with what a human would define as the concept. There may be other ways in which these model-derived concepts are subtly different from a concept that a human would request to be unlearned. This is fairly speculative, however. The authors do seem to acknowledge limitations in deriving concepts directly from the model.
7.	Results are generally not as convincing with OLMo as Llama-2 (for example Figure 4). Submissions would benefit from evaluating another model.
8.	Use of Needle as an 'oracle' is called into question in Figure 3, since for OLMo Needle performs very similarly to other methods, and appears to even be outperformed by MEMIT. This could indicate issues with the methodology (e.g. for validating concept vectors).

[1] Li, Nathaniel, et al. "The wmdp benchmark: Measuring and reducing malicious use with unlearning." arXiv preprint arXiv:2403.03218 (2024).

**Questions:**

1.	Suggestion: reword claims on novelty of internal evaluation for unlearning.
2.	Question: can you explain your reasoning for extracting potential concept vectors in this manner as opposed to others?
3.	Question: could you address some of the concerns around the selection of concept vectors, especially their utility and cases such as the McDonald's concept?
4.	Question: can you explain why results for Llama and OLMo are different?
5.	Question: can you explain the poor performance of Needle in Figure 3 (and 5 and 6)?

---

> ### Author Response · Authors · 2024-11-21
> **Responses to the Reviewer hnu6 (1/2)**
>
> We thank the reviewer hnu6 for putting an effort into reviewing our work. The reviewer has raised several important points and questions, which we clarify below.
>
> > **W1 (Previous intrinsic attempts):**
>
> Thanks for referring us to previous relevant work. We believe there is a fundamental difference between predicting knowledge based on internal representations [1, 2]---a predictive task---and between identifying specific parameters (in the form of “concept vectors” that constitute a small fraction of the model’s parameters) whose ablation behaviorally changes the predictions of the model.
>
> > **W2 (Vectors in the standard basis):**
>
> We acknowledge the fact that concepts, and knowledge, are not necessarily encoded in the standard basis (axes, or neurons). Following previous work on parametric encoding of knowledge, we search for axis-aligned concept vectors just because they are easier to find, as the search problem becomes more tractable. We agree that there may exist additional, non axis-line concept vectors. Yet, the vectors we do find are a **lower bound** on the amount of intrinsic knowledge still encoded in the model after unlearning: we show that these axis-aligned vectors exist, and are behaviorally relevant.  Therefore, we posit that removal of the concept knowledge encoded in these vectors is a necessary condition for unlearning. Our results show that current fine-tuning-based unlearning methods largely fail to introduce meaningful changes to these parameters.
>
> > **W3 (Manual work in Dataset generation):**
>
> In (a), a “clear pattern” refers to cases where at least 40% of the top-k tokens in the projection are keywords related to the target concept. A “concrete and specific concept” refers to a concept that is not a broad category that potentially covers many entities, such as “U.S. Presidents,” but rather a specific entity, such as “Barack Obama” or “Harry Potter.” In (b), we ensure that every related question is a single, straightforward query without complex contexts. Additionally, these questions are designed to be easily answerable by humans with knowledge of the target concept.
>
> > **W4 (Identifying concept vectors with specific tokens):**
>
> Similar to the point above (W2), we note that this search method allows us to at least identify this set of behaviorally relevant concept vectors. We agree that there may exist more.
>
> > **W5:** "*It is unclear what quality-control measures... is a place where things could go wrong.*"
>
> We are not fully understand the reviewer’s concern, and would like to ask for a clarification. We try to answer based on our understanding: When using the tokens obtained from the concept vector projection to search for and match the corresponding concepts, as described in Section 3.1, we first use GPT-4 to generate potential themes or concepts. These suggestions are then manually verified and filtered. After this, we conduct **causal validation experiments** to verify the matching between the selected vector and the target concept. This step is crucial to ensure that the concept vector aligns with our defined concept, as well as with the corresponding QA pairs and relevant text segments from Wikipedia used in the behavior tests. In this process, we manually verify that each concept’s QA pairs and corresponding Wikipedia text segments are both appropriately matched and reasonable. More details about the quality verification of generated behaviors tests can be found in Section 3.3.
>
> > **W6:** “*taking concepts directly…subtly different from a concept that a human would request to be unlearned*”:
>
> The goal of this work is to pose an evaluation test for unlearning. While the concepts selected were derived from the model, they are grounded in concrete concepts in the world, with corresponding Wikipedia pages describing knowledge about them. Moreover, similar concepts (from Wikipedia) are commonly studied in the context of knowledge removal and editing. Further, our experiments validate that this vector is causally related to the human-defined concept in terms of model behavior and also has specificity (Section 3 and Figure 4). We have also demonstrated that the corresponding concept vector is highly activated and invoked during the model's usage of knowledge associated with the human-defined concept (Section 5.1 and Table 4), further confirming that this concept vector contains at least a portion of the key target knowledge related to the human-defined concept. Taken together, we do not see why the fact that the concepts were derived from the model itself, given the goal of the paper, the experimental results, and relation to prior works, is a weakness.

---

> ### Author Response · Authors · 2024-11-21
> **Responses to the Reviewer hnu6 (2/2)**
>
> > **W7&W8 (Needle vs. other methods):**
>
> We believe the reviewer may not understand the interpretation of these results and would like to elaborate. First, note that Needle is not an unlearning method advocated in this paper. Instead, it serves as an oracle baseline, which has access to some of the parameters encoding the concept knowledge. Regarding its performance, we consider the effect that Needle introduces substantial for several reasons:
> - Needle only ablates **a single vector**, which constitutes **<0.001%** of the network parameters. It is much more “surgical” and fine-grained than other examined methods: MEMIT modifies a total of **44K** MLP vectors (44K times more vectors than Needle), **including the concept vector itself**, thus its better performance is quite expected. Therefore, we consider the unlearning effect that Needle introduces to the model’s ability of generating concept knowledge is already dramatic, showing that it does capture essential knowledge about the concept. As a reference, ablating a random vector in MLP layers keeps a BLEU score of **1.0**.
>
> - The distribution of knowledge within the model is broad and dispersed. In Needle, we only ablate one of the most prominent vectors related to the target concept. Furthermore, in this baseline, we applied simple noise perturbation and fine-tuning adjustments, without introducing any additional operations.
>
> **Regarding the reviewers' questions:**
>
> **Q1:** Please see our response in W1 and the response to the 9s3u’s Q4. We believe that our work is advanced and pioneering, as we approach the study of unlearning effects from the perspective of knowledge in model parameters, investigating whether it is influenced by existing unlearning methods. Previous works have primarily focused on the representation space, including disrupting activations to influence the expression of target knowledge in internal representations [1]. However, they overlooked the consideration from the model parameter perspective and the method proposed has not genuinely impacted the knowledge stored in the parameters.
>
> **Q2:** Please see our response in W2. Through the vocabulary projections and causal validation experiments in behaviors testing, we can ensure that the concept vector we identified is causally related to the model’s generation behavior about the target concept. Additionally, it is proved to be highly activated when the model processes relevant knowledge to answer related questions (Table 4 in Section 5.1).
>
>
> **Q3:** Please see our response in W4 and W5.
>
>
> **Q4:** (1) Compared to LLaMA2-7B-chat, OLMo-7B is a more fragile model, making it more prone to breakdown during finetuning-based unlearning. As a result, under the same conditions (e.g., epochs, learning rate, etc.) of finetuning unlearning, OLMo's general capabilities deteriorate more quickly. (2) Based on our previous experiments, we found that knowledge in OLMo is distributed more dispersed than in LLaMA. This is one of the key reasons why ablating a single concept vector (Needle) in OLMo is less effective than in LLaMA. This is an interesting issue that warrants further exploration in future studies.
>
> **Q5:** Please see our response in W7&W8.
>
> ---
> **References:**
>
> [1] The wmdp benchmark: Measuring and reducing malicious use with unlearning. ICML 2024
>
> [2] Estimating Knowledge in Large Language Models Without Generating a Single Token. EMNLP 2024

---

> > ### Comment · Reviewer_hnu6 · 2024-11-25
> > **Response Follow-Up**
> >
> > Thank you for the detailed responses.
> >
> > Re: W1, I agree that they are different, however I think the existence of that benchmark limits the claim that this is “the first benchmark for internal evaluation of unlearning methods.”
> >
> > Re: W3-5, These highlighted a general concern that working with concepts is difficult because it is hard to quantify what defines a concept, and thus including specific details about your process for selecting concepts is important (so that it can be replicated, for example). For example, adding the details you included in your response to W3 would be helpful. For a specific example of why I have some concerns: the ‘McDonalds’ concept in Table 7 for ChatGLM3-6B appears to be polysemantic: both a ‘McDonalds’ concept, but also a ‘names that start with Mc’ concept (McC, McM, McK, McD, McL). Could you maybe address that specific example, and argue why this is an outlier?
> >
> > Re: Q1, Will you include the RMU results in the paper? That would be an improvement.
> >
> > Re Q4, It is interesting that the behavior of OLMO is so different. It would be good to include some discussion of that in the paper. Additionally, as noted in W7, it would be ideal to evaluate another model to understand which model is an outlier.

---

> ### Author Response · Authors · 2024-11-29
> **Further Clarification for Reviewer hnu6**
>
> Thanks, Reviewer hnu6, for providing valuable feedback and questions.
>
> > *however I think the existence of that benchmark limits the claim that this is “the first benchmark for internal evaluation of unlearning methods.”*
>
> Thanks for your suggestion. We have already updated the claim in the introduction in Lines 043 - 044 from "*This work presents the first benchmark for internal evaluation of unlearning methods.*"  to  "*This work presents the first benchmark for **parameter-based** internal evaluation of unlearning methods*"
>
> **For more additional experiments we have conducted and further revisions we made to the paper, please feel free to refer to our General Response to all the reviewers we just posted.**
>
> > Will you include the RMU results in the paper?
>
> Yes, we have already included the RMU experiments and results in our submission’s pdf. Please see Table 3 in Section 4, Figure 3 in Section 5, Figure 5 and Figure 6 in the appendix of the updated PDF for the results of RMU. We also pointed out the RMU limitations in Lines 321 - 328 and Lines 374 - 376.
>
> From the experiments, we demonstrate that RMU does not achieve the same level of unlearning effectiveness as Model Editing. This is because its loss function is designed to disrupt the model's activation on the target knowledge parameters, rather than truly removing the target knowledge within them.
>
> > and thus including specific details about your process for selecting concepts is important (so that it can be replicated, for example). For example, adding the details you included in your response to W3 would be helpful. For a specific example of why I have some concerns: the ‘McDonalds’ concept in Table 7 for ChatGLM3-6B appears to be polysemantic: both a ‘McDonalds’ concept, but also a ‘names that start with Mc’ concept (McC, McM, McK, McD, McL). Could you maybe address that specific example, and argue why this is an outlier?
>
> During the process of collecting concept vector candidates and performing vocabulary projections, we often find some concept vectors tend to be intertwined with both lexical and semantic associations. Therefore, we believe it is normal to observe that example whose projection has some tokens with **prefixes** starting with "MC" (e.g., McC, McM, McK, McD) as well as tokens **semantically related** to "McDonald's" (such as Fast, Burger, or Mac, as in Big Mac).
>
> However, to truly confirm the causal relationship between the parameter vector and the defined concept, we rely on the Causal Validation Experiments described in Step 3 of Section 3. For the examples provided in Table 7, we have experimentally validated the causal connection between these parameters and the target concept knowledge in terms of model behavior.
>
> > *It would be good to include some discussion of that in the paper.*
>
> Thanks for the suggestion. We have mentioned that OLMo-7B is a more fragile model in Lines 1164 -1165 to indicate that it requires a smaller learning rate for fine-tuning compared with LLaMA2-7B-chat.
>
> And we will include more discussions about “*why the behavior of OLMO in Needle is so different*” in our paper after we conduct further and more precise experiments to verify that knowledge in OLMo is more dispersed than in LLaMA. This will help explain why ablating a single concept vector (Needle) in OLMo is less effective than in LLaMA. However, at this stage, this topic **slightly exceeds** the scope of the current paper, and the main contributions and findings presented so far are already clear and distinct.
>
>
> > *Additionally, as noted in W7, it would be ideal to evaluate another model to understand which model is an outlier.*
>
> Due to the limited time during the rebuttal period, it is challenging for us to rerun the complete ConceptVectors experiments on the third model.
>
> However, we believe that the consistent findings from our experiments on two transformer-based models, LLaMA and OLMo, and the concept vectors with strong causal effects that we have identified in other transformer-based models (such as Qwen, Mistral, and Chat-GLM) in Table 7 have provided compelling evidence and greatly support the validity of our experimental findings. We also have provided an automated pipeline for people to reproduce a new ConceptVectors benchmark on any other transformer-based model to reproduce the findings in Section A.7.
>
> We would greatly appreciate it if our revisions and additional experiments conducted in the updated paper could address the reviewer's concerns or doubts.

---

### Official Review · Reviewer_6fAd · 2024-11-03

**Soundness:** 2
**Presentation:** 2
**Contribution:** 2
**Rating:** 5
**Confidence:** 4

**Summary:**

This paper introduces a benchmark for unlearning methods to test residual intrinsic knowledge (i.e. stored in model parameters) and evaluate if respective unlearning methods _causally_ erase the concepts to be unlearned. This benchmark is motivated that current unlearning methods only unlearn in a shallow way (i.e. behavior) while the model continues to possess conceptual knowledge that can be adversarially retrieved (e.g. by using an adversarial suffix, advanced prompting, etc.). Towards this, they introduce a method, _Needle_, to identify "concept vectors" within transformer models' MLP layers -- that encode the said concepts. Using this method, this work introduces a metric to enumerate how much a given unlearning technique is _causally_ erasing the underlying knowledge. The paper presents experimental results that [i] include assessing various unlearning methods with their _intrinsic_ metric + targeted unlearning behavioral metrics i.e.  (Table 3) and [ii] the impact of adversarial attacks to extract the underlying unlearned knowledge on various respective unlearning methods including their own introduced method _Needle_ (Figure 3).

**Strengths:**

- The draft is well-written, presentable, and is read easily.
- The results from (ii) [from summary] are strong. The introduced method affects untargeted QA less than baselines when prompted adversarially (Figure 3).

**Weaknesses:**

The main weakness that makes the work uncompelling is that it breaks the hypothesis that "concept vectors" exist by definition. Unlearning at its core is supposed to return a model that operates like it has never seen a forget set which is primarily _behavioral change_ (while preserving the utility of the model as much as possible). I agree with the authors that this is not causal since information is still retrievable through advanced prompting or adversarial attacks, alluding to the fact that residual knowledge is retained. But given the fact that Needle (concept vector perturbation) itself doesn't induce such behavior change (getting 3x worse bleu score than the simple gradient ascent in Table 3) points the concept vectors don't work as hypothesized/defined and are thus not suitable for evaluating/benchmarking unlearning. I would argue that claiming to measure _causal_ unlearning, (and not just behavior) is even more strict and doesn't make sense if it's not backed up by behavior unlearning ( in fact, it's not causal at all in such case).

This gap in performance might be due to the violation of the unsaid (and/or untested) assumption that information or concepts reside in a concentrated manner in a few selected params and are not diffused across params globally. It would be great to have this assumption experimentally validated. In other words, intrinsic should at least rank order behavioral scores which it doesn't.

Other than that, the reported results include manually reviewing/handpicking concept vectors -- which would be not scalable to benchmark unlearning methods, making it impractical.

The adversarial attacks used for experiments (Section 5) do not include the strongest attack such as GCG [1].

It would be better to see the model evaluation harness scores [2] than just untargeted QA to see the degree to which Needle affects the underlying model utilities. Also, it would be better to see the performance (behavior vs model utility, untargeted QA) by individual adversarial methods than aggregated over multiple methods like in Figure 3.

The current set of experiments doesn't dig deeper to be convincing in my opinion. For example, if we look at the _concept vectors_ from a model trained without _forget set_ (samples provided for unlearning) i.e. a control model and keep other training data/scheduling as is (i.e. in controlled settings), we can compare that to validate if the knowledge is in fact concentrated in said vectors. This is computationally difficult for larger models, but certainly possible for smaller GPT-2 level models as done in [3].

---
[1] Andy Zou, Zifan Wang, J. Zico Kolter, and Matt Fredrikson. Universal and Transferable Adversarial Attacks on Aligned Language Models.

[2] Gao, L., et al, "A framework for few-shot language model evaluation," 2024, https://github.com/EleutherAI/lm-evaluation-harness

[3] https://physics.allen-zhu.com/

**Questions:**

1. In line 202, "Note that in both settings there is no need for gold answers or references, as our goal is to evaluate the effect of unlearning on the model’s outputs" -- why don't we need gold answers?
2. In Step 3, line 207, how do you ensure that these concept vectors aren't cohesively tied up with other concepts? Like parameters are compression of training data -- so they would likely to be not isolated to selected concepts.

---

> ### Author Response · Authors · 2024-11-21
> **Responses to the Reviewer 6fAd  (1/2)**
>
> We thank the reviewer for their thorough review and constructive comments. We wish to address the reviewer’s points and hope to engage in discussion.
>
> > **W1:** “*given the fact that Needle (...) itself doesn't induce such behavior change… points the concept vectors don't work as hypothesized/defined and are thus not suitable for evaluating/benchmarking unlearning*”:
>
> We interpret these results differently, consider the effect that Needle introduces substantial, and strongly believe that concept vectors are valuable for evaluation. Table 3 shows that Needle obtains mean target bleu scores of 0.46 and 0.31 for the two models. These are indeed higher (worse) than the other unlearning methods (which get to 0.05-0.24). However, there are several important things to keep in mind:
>
> - Most importantly, as illustrated in Figure 3, the concept vectors we identify suggest the presence of residual knowledge traces, which are eventually behaviorally relevant: they make the model more vulnerable to jailbreaking attacks. While standard behavioural evaluations do not reveal the importance of these concept vectors, a deeper robustness analysis through jailbreaking reveals their **complementary** role to standard behavioural evaluation.
>
> - We do not expect that the concept vector is the **only** parameters in the network that capture knowledge about the concept. Previous works show that knowledge is spread across multiple places in the network [1,2], so overall it is expected that ablating a single vector would not erase the knowledge completely. Still, even though the concept vector is not necessarily the only place where knowledge is stored, it is still useful for evaluating unlearning – given its causal effect, as described above.
>
> - Needle only ablates **a single vector**, which constitutes **<0.001%** of the network parameters. We consider the effect it introduces dramatic, as ablating a random vector in MLP layers will keep a BLEU score of **1.0**. A decrease of BLEU from 1 to <0.5 means that this vector plays a causal (and specific) role in the model’s generation of knowledge about the concept. Other methods obtain better scores but they are either not restricted and modify all the parameters or modify the MLP layers where the concept vectors are in. For example, MEMIT modifies a total of **44K** MLP vectors (44K times more vectors than Needle), including the concept vector itself.
>
> > **W2:** Regarding the manual selection of concept vectors:
>
> The data selection process is mostly automated and systematic (which was appreciated by Reviewer 9s3u). For creating a high-quality benchmark, there is indeed a manual selection phase, which finds the vectors that correspond to the most coherent concepts based on human judgment, followed by a causal validation step.
>
> Please note that we have included another more automated version of our pipeline which is easier to scale in Section A.6 in the appendix, and can be used to find concept vectors in any transformers-based models. The whole process of adding one new concept’s data takes no more than 90 seconds.
>
> > **W3:** Regarding the GCG adversarial attack:
>
> Thanks for this note. Following the reviewer’s comment, we additionally provide the effects of two current representative Jailbreak methods – GCG [3] and AutoDAN [4] – on the activation enhancement of concept vectors, and compare them with the effects without jailbreak and with the first crafted prompt in the paper. We observe similar trends, where jailbreak enhances the activation of model knowledge parameters.
> We will add these results to the paper.
>
> | Model / Attack | No Jailbreak | Crafted Prompt1 | GCG | AutoDAN |
> |-------|-------|-------|-------|-------|
> | Unlearned via Gradient Difference | 2.14 | 3.07 | 3.51 | 3.20 |
> | Unlearned via DPO | 1.42 | 2.03 | 2.92 | 2.65 |
> | Vanilla | 2.59 | 3.34 | 4.02 | 3.84 |
>
> > **W4:** Regarding the unrelated abilities evaluation:
>
> Thanks for your suggestions. Notably, while evaluating the effect of unlearning on general capabilities of the model is valuable, it is tangential to the main goal of the benchmark – evaluating unlearning of knowledge, which we address from multiple angles. Specifically, the evaluation on unrelated questions does provide a signal regarding the general generation abilities of the model beyond the unlearned concept. Nonetheless, we see the value in the reviewer’s suggestion and are now working on adding the harness evaluation (given that our submission received 7 reviews, we may not finish these experiments on time for the rebuttal).

---

> > ### Author Response · Authors · 2024-11-21
> > **Responses to the Reviewer 6fAd (2/2)**
> >
> > > **W5:** Regarding the performance of individual adversarial methods:
> >
> > Please note that our primary focus is not comparing the advantages of individual adversarial attack methods, but rather examining the performance and limitations of the existing unlearning baselines under adversarial attacks. Therefore, while we agree that such experiments could be indeed interesting, they are not vital for supporting the arguments made in this work and so we leave them for future work to explore, while providing our benchmark to facilitate that.
> >
> > **Regarding the Questions:**
> >
> > **Q1:** We don’t need gold answers because: (1) We directly evaluate the effect of unlearning by comparing the model’s original output with its output after undergoing unlearning training.
> > (2) During the validation and filtering stages of the benchmark, we have ensured that the model’s original output accurately reflected the correct answers to the questions.
> >
> > **Q2:** As stated in our limitations part (Section 7), we acknowledge that concept vectors are often exhibit superposition with other knowledge. However, this does not affect our evaluation process. We believe that modifying these vectors, which contain critical target knowledge, is at least a necessary condition for unlearning. From the results, it is evident that current fine-tuning-based unlearning methods have largely failed to modify the parameters storing the target knowledge and have not disrupted its storage.
> >
> > ---
> > **References**:
> >
> > [1] Knowledge Neurons in Pretrained Transformers. ACL 2022.
> >
> > [2] Dissecting Recall of Factual Associations in Auto-Regressive Language Models. EMNLP 2023.
> >
> > [3] Universal and Transferable Adversarial Attacks on Aligned Language Models
> >
> > [4] AutoDAN: Generating Stealthy Jailbreak Prompts on Aligned Large Language Models. ICLR 2024

---

> > > ### Comment · Reviewer_6fAd · 2024-11-26
> > >
> > > Thanks for the response and addressing part of my concerns. However, I'm still dissatisfied that Concept Vectors cannot be used to benchmark unlearning methods since behavioral change is not observed (Table 3). Labeling this _causal_ is problematic and doesn't agree with the presented results. Moreover, the table in the W3 response doesn't contain _Needle_ method. In my judgment, it is uncertain to which degree concepts are embedded/concentrated into _concept vectors_ lack proper understanding around them. I will stand with my current score. I will keenly read the responses through the rebuttal period and offer responses wherever I can.

---

> ### Author Response · Authors · 2024-11-29
> **Further Clarification for Reviewer 6fAd (1/2)**
>
> Thanks, Reviewer 6fAd, for your valuable suggestions and questions regarding our paper.
>
> > *I'm still dissatisfied that Concept Vectors cannot be used to benchmark unlearning methods since behavioral change is not observed (Table 3). Labeling this causal is problematic and doesn't agree with the presented results.*
>
> Please refer to the updates we have made in **Table 3** in the updated PDF, as well as the new discussion provided in **Lines 356 - 370**. (For a summary, it can also be seen in the General Response message we just posted.)
>
> Specifically, we updated the Needle method’s performance in Table 3 by removing its gradient ascent fine-tuning step. We found that this step caused further degradation of the model's unrelated knowledge without significantly improving unlearning of the target knowledge.
>
> After removing this step:
>
> - The updated Needle achieves the **best trade-off between unlearning target knowledge and preserving unrelated knowledge, compared to all other unlearning baselines**. It shows the **largest difference in behavior metrics** between Target QA scores and Unrelated QA scores: on LLaMA BLEU and ROUGE-L: 0.415 | 0.301, and on OLMo BLEU and ROUGE-L: 0.334 | 0.455.
> - Needle can introduce a clear causal effect of an average decrease of >0.5 BLEU points in the behavioral tests. This effect is by definition causal as Needle intervenes in the computation and consequently modifies the ability of the model to answer questions about the target concept.
> - Additionally, across both models, Needle maintains the **highest Unrelated scores** among all baselines, indicating that it has minimal impact on the model's unrelated knowledge.
>
> These suggest that the concept vectors we identified are strongly causally related to the target concept, and we respectfully do not consider this causal relationship to be weak.
>
> For a detailed view for the **specific distributions** of the strength of the behavioral causal relationship between each vector and the target concept, please refer to **Figure 4**.
>
> In the jailbreak experiments (Section 5), we also confirm that both manual jailbreak prompts and automated jailbreak methods (GCG, AutoDAN) significantly increase the activation of these identified concept vectors, while the activation of unrelated vectors remains very low ([-0.002, 0.003]). **This is a clear and observable mechanism** (Table 4, Figure 7 and Figure 8).
>
> > *Moreover, the table in the W3 response doesn't contain Needle method.*
>
> Since Needle only modifies the specific concept vector for each edit and does not update the other parameters, it will not change the activations on the certain concept vectors. Therefore, the activations of the target concept vectors will **remain identical to those observed in the vanilla model**.
>
> Methods like DPO and gradient difference perform full parameter fine-tuning, so they will affect the activations on the concept vectors.
>
> Please note that the "activations" referred to here are defined in Lines 421 - 434, which means the **coefficient scores** assigned to each dimension of the concept vectors (See the formula in Lines 110 - 114).

---

> ### Author Response · Authors · 2024-11-29
> **Further Clarification for Reviewer 6fAd (2/2)**
>
> > *In my judgment, it is uncertain to which degree concepts are embedded/concentrated into concept vectors lack proper understanding around them*.
>
> Please refer to Figure 4 in the Appendix A.4 if you would like to have a more intuitive understanding of the degree to which each concept vector is behaviorally linked to its corresponding target concept.
>
> **While in this paper we did not separately analyze the depth of knowledge encoded in each concept vector, but rather treated them as a whole by averaging**, based on the ConceptVectors benchmark we have developed and the experiments we have conducted, our work **have been able to provide several authentic and important** findings and contributions, including but not limited to the following:
>
> - Finetuning-based unlearning has very limited effect on the parameters storing the target knowledge, and it does not truly erase the model's target knowledge (Table 3).
> - Finetuning-based unlearning will inevitably impacts the other unrelated knowledge behavivors and is not suitable for true unlearning in LLMs. (Table 3, Figure 3, Lines 465 - 469)
> - Finetuning-based unlearning will include suppressing the activations on the model's target knowledge parameters to help it achieve the unlearning effect on behaviors metrics, even though it do not truly erase the knowledge. (Table 4, Lines 448 - 449)
> - Model editing-based approaches, including MEMIT and Needle, are capable of making more targeted parameter modifications to the target concept vectors without affecting unrelated knowledge. As a result, they achieve stronger unlearning effectiveness (Lower Target Knowledge Scores in Table 3), Specifictiy  (Higher Unrelated Knowledge Scores in Table 3), and Robustness to Jailbreak attacks (Figure 3, Figure 5, Figure 9).
> - Jailbreak significantly increases the activations of the target concept-related knowledge parameters (Table 4, Figure 7, and Figure 8) without affecting the activations of unrelated parameters.
>
> We believe that these contributions we have made are crucial and important for the future development of the Knowledge Unlearning field in transformers-based LLMs.
>
> And we would greatly appreciate it if this could help address the reviewer's concerns or doubts.

---

### Official Review · Reviewer_tU7V · 2024-11-04

**Soundness:** 3
**Presentation:** 2
**Contribution:** 3
**Rating:** 6
**Confidence:** 3

**Summary:**

The paper proposes an evaluation metric of LLM unlearning using vocabulary projections and concept vectors. The authors construct a benchmark dataset based on this, named CONCEPTVECTORS. In the paper, they conducted experiments to study the performance of various unlearning methods using this benchmark.

**Strengths:**

The paper provides a novel aspect to evaluate the LLM unlearning, using the encoded concept vectors. This leads to a more internal and intrinsic way to unlearn certain information. In this paper, they verified that the existing methods mostly only suppress the information, instead of erasing them. Then uses both intrinsic evaluation and behavioral tests to valid the idea. Combined with the jailbreak attacks, the experiments greatly improve the quality and significance of their work.

**Weaknesses:**

First, a relatively minor weakness is on the presentation of the paper. The discussion about the concepts is clear. However for the discussion of constructing the dataset. There might be too much detailed discussion about validations of manual quality checking by humans, it could possibly be moved into the Appendix. Besides that, there are many hyperparameters in the entire experiment setup and dataset construction. It would be better to provide a more systematic exploration to study their effects, in order to make the results and the contribution more robust. Moreover, the results seem to be limited to the two chosen models (LLaMA and OLMo).

**Questions:**

The concept vectors are based on the two models LLaMA and OLMo. For unlearning using other models, is there a way to directly apply the benchmark CONCEPTVECTORS to use on other model architecture?

---

> ### Author Response · Authors · 2024-11-21
> **Responses to the Reviewer tU7V**
>
> We thank the reviewer for their thoughtful review, finding our evaluation novel, and appreciating the experiments. We address the points raised:
>
> > **W1:** *“a relatively minor weakness is on the presentation of the paper... There might be too much detailed discussion about validations of manual quality checking by humans, it could possibly be moved into the Appendix”*
>
> Thank you for the suggestion. We will move unnecessary details to the appendix in the final version of this work.
>
> > **W2:** *“many hyperparameters... It would be better to provide a more systematic exploration to study their effects”*
>
> The paper has provided the analysis of hyperparameter selection:
> - For the noise scale used in both the causal validation experiment and the Needle method, please see Section A.3 in the appendix.
> - For the hyperparameters used in running the various unlearning baselines, please see Section E in the appendix.
> - For the data splits in our benchmark, please see Section 4.
>
> We are happy to add more analysis, if these do not cover what the reviewer is looking for.
>
> > **W3:** *“the results seem to be limited to the two chosen models”*
>
> Please note that the purpose of the benchmark is to evaluate the performance of different unlearning methods, rather than to evaluate model performance. Thus, we focus on covering a diverse set of unlearning methods rather than models, including 9 different baselines from 4 different categories. In addition, the overall approach of using concept vectors is generally applicable. Their existence has been shown in multiple transformer-based language models, including GPT2 [1,2,3] and GPT-J [3], and also in the appendix (see Table 7), where we also validated that concept vectors can be located in ChatGLM3 and Qwen1.5. Therefore, it should be easy for future work to extend this benchmark in case they are interested in unlearning for specific models.
>
>
> **Regarding the reviewers’ question:**
>
> **Q1:** Evaluation with ConceptVectors can be done on the two selected widely-used models. However, it can be extended to other models – for models that were derived from the selected models (i.e. fine-tuned versions of LLaMA and OLMo) it merely requires re-validating the concept vectors through the causal validation step in our pipeline, and for other models it requires locating other concept vectors (which we have demonstrated in the appendix – Table 7).
>
> ---
> **References**
>
> [1] Transformer Feed-Forward Layers Build Predictions by Promoting Concepts in the Vocabulary Space. EMNLP 2022.
>
> [2] Analyzing Transformers in Embedding Space. ACL 2023.
>
> [3] Dissecting Recall of Factual Associations in Auto-Regressive Language Models. EMNLP 2023.

---

> > ### Comment · Reviewer_tU7V · 2024-11-24
> >
> > Thanks to the authors for the responses. Since I have not seen the corresponding updates of the paper, and the explanations provided in the responses address my concerns only to a very limited extent. I will maintain the original overall score.

---

> > > ### Author Response · Authors · 2024-11-29
> > > **Further Clarification for the Reviewer tU7V about the Paper updates**
> > >
> > > Thank Reviewer tU7V for your continued consideration of our paper.
> > >
> > > We have taken your feedback into account and already made the corresponding revisions to the relevant sections of the updated paper:
> > >
> > > - Move the unnecessary details about datasets quality verification in Section 3.3 to the Appendix A.4.
> > > - For the hyperparameter settings exploration, we have updated Section E in the appendix to make its description clearer. Additionally, we have included the RMU method in the paper and provided its corresponding hyperparameter details.
> > > - For the other updations in our paper, please check our new General Response for more details.
> > >
> > > Thank you once again for the valuable reviews, and please feel free to raise any points that remain unclear.

---

### Author Response · Authors · 2024-11-29
**General Message to all the Reviewers about the Submission's pdf Revisions and the Additional Experiments conducted**

Thanks to all the reviewers for actively participating in our paper discussion and providing constructive suggestions and valuable feedback.

We have incorporated your insights and suggestions in the submission’s pdf, conducted additional experiments to strengthen our arguments, and revised the paper to improve its structure, making the presentation more coherent and precise.

**Below are the additional experiments we have completed:**


|Additional Experiments| Contributions & Findings | Updated Location | Related Reviewers |
| --- | --- | --- | --- |
| 1. Conducted additional experiments on the two automated jailbreak methods, **GCG** [1] and **AutoDAN** [2], confirming that they also increase activation of the target concept vectors. | This further reinforces the claim that jailbreak techniques can lead the model to enhance these target knowledge parameters to help bypass unlearning. | Please see Table 4 in the updated PDF. | Suggested by **Reviewer 6fAd** and **Reviewer 1G7R** |
| 2. Among the unlearning baselines, we newly included **RMU** [3], a representative method from the category of representation engineering, in both the main experiments and the jailbreak experiments. | We demonstrate that the representation engineering-based method, RMU, does not achieve the same level of unlearning effectiveness as Model Editing. This is because its loss function is designed to disrupt the model's activation on the target knowledge parameters, rather than truly removing the target knowledge within them. | Please see Table 3 in Section 4, Figure 3 in Section 5, Figure 5 and Figure 6 in the appendix of the updated PDF. We pointed out the RMU limitations in Lines 321 - 328 and Lines 374 - 376. | Suggested by **Reviewer hnu6** and **Reviewer 1G7R** |
| 3. We **updated the performance of the Needle method in Table 3** by revising its steps. Specifically, we removed the subsequent gradient ascent fine-tuning process in Needle, as it impacted the model's unrelated knowledge more significantly, and provided minor benefits in removing the target knowledge. | The updated Needle method significantly reduces the impact on unrelated knowledge, while maintaining the unlearning effect on the target knowledge largely unchanged. Compared to all other unlearning baselines, it **achieves the best trade-off between unlearning the target knowledge and preserving the unrelated knowledge in the behavior metrics**. It further supporting our argument that there is a strong causal relationship between the concept vectors we target and the corresponding concept knowledge. For an effective and truly thorough unlearning method, it is **crucial** and **required** to modify these concept vectors. | Pleae see **Table 3** in the updated PDF and the new discussion of it in **Lines 356 - 370**. | Solving the **Reviewer 9s3u's** and **Reviewer 6fAd's concerns** on Table 3. |


In addition to incorporating these additional experiments and their corresponding discussion into the paper, we have also made further revisions to the paper structure and wording based on your feedback to enhance its rigor.

**The revisions of our paper contains:**

- Move the unnecessary details about datasets quality verification in Section 3.3 to the Appendix A.4. (Suggested by the **Reviewer tU7V**)
- Update the claim in the introduction in Lines 043 - 044 from "*This work presents the first benchmark for internal evaluation of unlearning methods.*"  to  "*This work presents the first benchmark for **parameter-based** internal evaluation of unlearning methods*" (Suggested by the **Reviewer hnu6**).
- Describe the RMU Limitation and highlight its differences in Lines 321 - 328 and Lines 374 - 376. (suggested by **Reviewer 1G7R**)
- Add GCG and AutoDAN’s description, results and discussion into the paper in Lines 427 - 429, Table 4 and Lines 430 - 445.
- Update Needle performance and its new description in the paper Lines 356 - 370 and Lines 330 - 333 .
- Update the Table 3 results discussion in Lines 348 - 401.

Thanks once again for all the reviewers’ active participations in our paper discussion. Since our paper has received feedback from seven reviewers and receive multiple constructive suggestions, we have made every effort to incorporate all these suggestions by adding additional experiments and refining the paper to make it more comprehensive and rigorous.

**We would sincerely appreciate it if the reviewers could check whether our revisions have addressed their concerns.**


---

**References:**

[1] Universal and Transferable Adversarial Attacks on Aligned Language Models

[2] AutoDAN: Generating Stealthy Jailbreak Prompts on Aligned Large Language Models. ICLR 2024

[3] The wmdp benchmark: Measuring and reducing malicious use with unlearning. ICML 2024

---

### Author Response · Authors · 2024-12-02
**Kind Reminder for all reviewers to review our latest paper revision, additional experiments and the new responses provided.**

Dear Reviewers,

We sincerely thank you for your constructive feedback and valuable discussions.

As the discussion deadline is approaching, we would appreciate it if you could please review our updated submission's pdf and additional experiments conducted following your suggestions, and reconsider your evaluation in light of these substantial improvements. **We have summarized all the revisions in the last general message *“General Message to all the Reviewers about the Submission's pdf Revisions and the Additional Experiments conducted”***.

Thanks again for your time and consideration.

Authors

---

### Meta-Review · Area_Chair_8AXw · 2024-12-17

**Metareview:**

This paper proposes a new benchmark for LLM unlearning, with the idea of testing that information is no longer stored in the weights rather than just that a behavior is mitigated.  The reviewers were conflicted about accepting this paper.  I agree with points raised by Reviewer WwUF (who gives the paper a score of 3), as well as other reviewers, namely that concept vectors are not suitable for an unlearning benchmark since the causal link is weak.  Concept vectors make sense to use for methods, where the weak causal link doesn’t matter as long as performance goes up, but benchmarks are a different story.  As such, I recommend rejection.

**Additional Comments On Reviewer Discussion:**

The authors did engage with reviewers and had back and forth, but they ultimately were not able to convince several reviewers (who did engage with them).

---

### Decision · Program_Chairs · 2025-01-22

Reject